# Methanol fixation is the method of choice for droplet-based single-cell transcriptomics of neural cells

Ana Gutiérrez-Franco[1,2,5], Franz Ake[1,2,5], Mohamed N. Hassan [1,2], Natalie Chaves Cayuela [1,2], Loris Mularoni[2,3] & Mireya Plass [1,2,4✉]

The main critical step in single-cell transcriptomics is sample preparation. Several methods have been developed to preserve cells after dissociation to uncouple sample handling from library preparation. Yet, the suitability of these methods depends on the cell types to be processed. In this project, we perform a systematic comparison of preservation methods for droplet-based single-cell RNA-seq on neural and glial cells derived from induced pluripotent stem cells. Our results show that while DMSO provides the highest cell quality in terms of RNA molecules and genes detected per cell, it strongly affects the cellular composition and induces the expression of stress and apoptosis genes. In contrast, methanol fixed samples display a cellular composition similar to fresh samples and provide a good cell quality and little expression biases. Taken together, our results show that methanol fixation is the method of choice for performing droplet-based single-cell transcriptomics experiments on neural cell populations.

---

[1] Gene Regulation of Cell Identity, Regenerative Medicine Program, Bellvitge Institute for Biomedical Research (IDIBELL), L'Hospitalet del Llobregat, Barcelona, Spain. [2] Program for Advancing Clinical Translation of Regenerative Medicine of Catalonia, P-CMR[C], L'Hospitalet del Llobregat, Barcelona, Spain. [3] Regenerative Medicine Program, Bellvitge Institute for Biomedical Research (IDIBELL), L'Hospitalet del Llobregat, Barcelona, Spain. [4] Center for Networked Biomedical Research on Bioengineering, Biomaterials and Nanomedicine (CIBER-BBN), Madrid, Spain. [5] These authors contributed equally: Ana Gutiérrez-Franco, Franz Ake. ✉email: mplass@idibell.cat

Single-cell transcriptomics (scRNA-seq) methods have revolutionized the way we study in a high-throughput manner the expression of genes across individuals, tissues and even in disease[1–3]. Previously, studies were limited to identify changes in the expression levels of genes in bulk, that is, in the population of cells that composed a particular sample. Thus, these approaches mixed two different effects: changes in the cell composition of the sample of interest and changes in the expression of genes within individual cells. Now, scRNA-seq allows assessing these two effects independently and can detect both changes in the cellular composition[4,5] and the expression of genes in specific cell types[6,7].

Despite the popularity of scRNA-seq methods, there are still several technical challenges unsolved. For instance, the dissociation of the cells from a tissue and the obtention of a good cell suspension, necessary for scRNA-seq, is highly tissue-specific and may require the use of different strategies including enzymatic digestion, mechanical disgregation, fluorescence-activated cell sorting, and other technologies[8–12]. As a result, the preparation of samples for scRNA-seq can take several hours, which makes more convenient to process them at later time points. Apart from technical difficulties, cell preservation is also important if we need to decouple sample dissociation and processing for other reasons such as the shipment of samples to an external facility, or if we want to collect multiple samples and process them together at a later time point to save time or money. In all these cases, researchers would like to preserve these samples in a way that minimizes the differences in cell composition and gene expression of individual cells in comparison to the original sample. That is, the best preservation method will be the one that has the smallest impact in the cell composition of the sample and the transcriptomic profile of the individual cells.

Several cell preservation methods have already been developed to overcome this problem and uncouple sample handling from library preparation. Among them, we find both home-made and commercial solutions including methanol fixation[13,14], dithio-bis succinimidyl propionate[15], dimethyl sulfoxide (DMSO) cryopreservation[16,17], acetic-methanol (ACME)[10], paraformaldehyde[18], CellCover[17], and vivoPHIX[19]. These methods aim at maintaining sample composition and RNA quality of cells. Yet, considering that sample preparation is tissue-specific, we expect that different preservation methods could be optimal for different samples. Many of these protocols have been tested only in cell lines or easy-to-obtain cells such as peripheral blood cells and thus, it is not clear how their performance is in cells that are difficult to dissociate or that are potentially damaged during dissociation. In particular, none of the previous methods have been tested in mature neurons or in human-induced pluripotent stem cell (hiPSC) derived neurons, which are the main source of neural cells for studying the molecular mechanisms driving neurological diseases[20].

In this work, we have compared the performance of five popular fixation or preservation methods in neural and glial cells derived from hiPSCs. The results from our work show that the different preservation/fixation methods affect the samples in different ways, including biases in the transcriptomic profile, cell composition and library complexity. DMSO cryopreservation provides the highest cell quality in terms of library complexity. Yet, the obtained datasets are strongly depleted of neurons and display a stronger stress signature. In contrast, ACME and vivoPHIX do not significantly affect the cell composition of the single-cell suspensions but damage the RNA, which reduces the library complexity and thus the number of genes and RNA molecules detected in individual cells. Taken together, our results show that methanol fixation is the method of choice for performing droplet-based single-cell transcriptomics experiments on neural cell populations as it provides a high library complexity without affecting the cell composition nor gene expression in comparison to fresh samples.

## Results

**Experimental setup for the systematic comparison of preservation protocols.** Individual or pooled hiPSC cell lines were differentiated to cortical neurons using a previously described protocol[21] with minor modifications. Briefly, hiPSC colonies were seeded in 12-well plates coated with matrigel, and after reaching confluency, neural differentiation was induced using a combination of BMP inhibitors (noggin, dorsomorphin, and SB431542) (Fig. 1a). After 29–50 days of differentiation, cells were dissociated with papain-accutase solution to obtain a single-cell suspension (Supplementary Data 1). At this point, some of the samples were directly encapsulated using Dolomite Bio automatic Drop-seq setup NADIA (fresh), cryopreserved with DMSO[17], or preserved using methanol[14,22], ACME[10], or chemical components that stabilize RNA molecules such as vivoPHIX[19] and CellCover[17], which have been previously used in single-cell transcriptomics (Fig. 1b).

**Cell preservation methods can impact RNA quality.** Before performing single-cell transcriptomics analysis, we investigated if the preservation of single-cell suspensions affects the quality of obtained RNA. For that purpose, we extracted total RNA from fresh and preserved neural precursor cells (NPCs) stored at −80 °C or 4 °C for up to 15 days. For each of the samples, we quantified the total amount of RNA and assessed its quality using the Agilent TapeStation system. Our results showed that the quality of the RNA extracted depended on the preservation method used. DMSO, methanol, and ACME samples had very high RNA integrity number (RIN) values (~9), similar to those of fresh samples. In contrast, the vivoPHIX sample had some RNA degradation (RIN ~7) and the samples processed with CellCover had stronger degradation levels, with RIN values ~2 at 4 °C and ~6 at −80 °C (Supplementary Fig. 1). These results demonstrate that CellCover is not suitable for long-term storage of cells for single-cell transcriptomics. Taking this into consideration, we decided to discard CellCover for the systematic comparison of preservation methods.

**Cell preservation affects the complexity of captured single-cell transcriptomes.** To compare the impact of different preservation methods, hiPSCs were differentiated to NPCs and either preserved using one of the different preservation/fixation methods or directly encapsulated with the NADIA equipment, a commercial Drop-seq setup (Fig. 1b). Cell encapsulation and library preparation was done following the same protocol for all samples. The obtained datasets were then evaluated using different metrics to assess their quality.

Inspection of the cDNA profiles showed that ACME and vivoPHIX samples had less cDNA and smaller fragments than the rest of the libraries (Fig. 2a), which is consistent with RNA degradation. This was expected in the vivoPHIX samples, which showed already a lower RNA quality after being preserved for 2 weeks, but not for ACME samples, which had RIN values similar to that of fresh samples (Supplementary Fig. 1). Next, we evaluated the impact of the preservation methods on library complexity. We used samtools[23] to downsample each of the unaligned BAM files to produce files containing 10%, 20%, 30%, etc. of the original dataset. Each of these subsamples was then processed using the same computational pipeline to generate downsampled digital gene expression (DGE) matrices. Our results show that at the same sequencing depth, DMSO

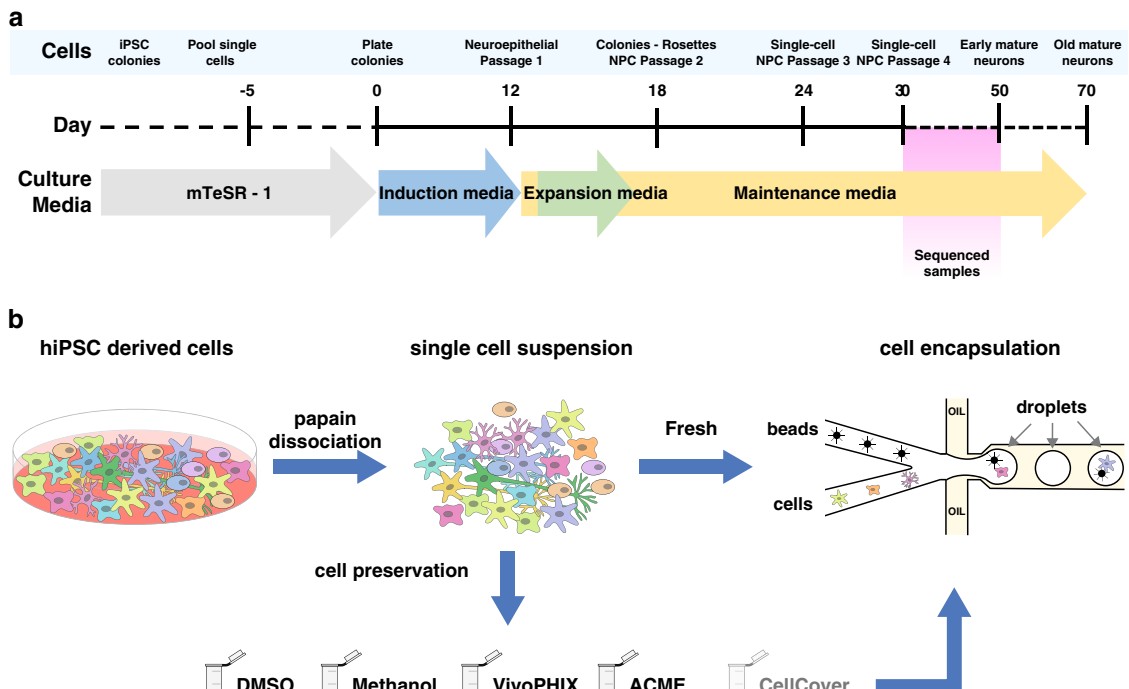

**Fig. 1 Experimental setup. a** Schematic representation of the differentiation protocol of hiPSCs to neural progenitor cells. **b** Systematic comparison of preservation methods. After the differentiation of hiPSCs, all cells were dissociated using papain-accutase and either directly encapsulated or preserved using one of the different reagents tested. After being preserved, cells were thawed or rehydrated and encapsulated using a commercial Drop-seq setup using the same protocols.

cryopreserved cells and methanol-fixed cells yield a comparable or higher number of genes and unique molecular identifiers (UMIs) per cell than fresh cells. In comparison, at 7500 reads per cell, ACME and vivoPHIX samples have around 40% of the Genes and UMIs obtained in fresh cells (Fig. 2b) (Supplementary Data 2). The higher number of genes and UMIs in the DMSO samples D1 and D2 and the methanol-fixed samples M3 and M4 is due to differences in the capture efficiency of the beads used for the encapsulation and not to an intrinsic higher capture efficiency in these experimental conditions (Supplementary Fig. 2 and Supplementary Data 1).

The lower complexity of vivoPHIX and ACME libraries is also reflected in the proportion of low-quality cells in the samples, which have few UMIs and genes detected per cell (Fig. 2c and Table 1), and may correspond to empty droplets or droplets containing broken cells[24]. While the number of low-quality cells discarded is around 10% of cells in fresh, DMSO, and methanol-fixed cells, this number increases up to 29% and 49% in ACME and vivoPHIX samples, respectively (Table 1 and Supplementary Data 3).

After discarding low-quality cells, the remaining vivoPHIX and ACME cells still show clear differences in quality (Fig. 2c and Table 1). Interestingly, the percentage of UMIs mapped to mitochondrial genes was similarly low across all samples, suggesting that the lower number of captured RNAs in these samples may not be due to RNA leakage or cellular damage[24] but rather be related with RNA capture efficiency or RNA degradation. In addition, ACME samples showed a much higher fraction of UMIs mapped to ribosomal proteins than any of the other samples (Fig. 2c), which has also been previously associated to low-quality cells or technical artifacts[25].

One of the reasons why vivoPHIX and ACME fixed cells have less RNAs per cell could be that the fixation method facilitates cell breakage. In this case, we would expect lower library complexity

and a higher fraction of reads coming from intronic regions, which mainly come from pre-mRNAs and are enriched in nuclei[26,27]. In our samples, the fraction of intronic reads coming from each sample was variable across preservation methods, ranging from 10% in DMSO samples to ~30% in methanol and vivoPHIX (Fig. 2d) samples. The higher fraction of intronic reads in vivoPHIX and methanol samples is indicative of nuclear RNA enrichment[26,27]. Yet, considering that the number of genes and UMIs detected in methanol is, on average, around three times higher than in vivoPHIX samples (Supplementary Data 2), RNA leakage is likely not the cause of this bias. Together, these results demonstrate that vivoPHIX and ACME preservation methods significantly affect the quality of the single-cell transcriptomes obtained from human NPC populations although this cannot be explained by an increased RNA leakage or cell breakage.

**DMSO cryopreservation alters the cell composition of hiPSC-derived cell populations.** After assessing the overall quality metrics of the samples, we investigated whether preservation methods affect the cellular composition of the samples. For that purpose, we pooled all the samples and analyzed them together. Initial analysis showed a strong batch effect driven mainly by the preservation method used. As can be seen in Supplementary Fig. 3, before integration, the cells from different experiments occupy different regions of the uniform manifold approximation and projection (UMAP) plot (Supplementary Fig. 3). Thus, we used Harmony[28] to integrate the datasets and identify the cell populations obtained from the hiPSC differentiation (Fig. 3). After batch correction, we identified 12 cell populations corresponding to proliferating progenitors, NPCs, astroglial precursors, intermediate progenitors and different types of neurons characterized by the expression of specific marker genes (Fig. 3, Supplementary Figs. 4 and 5, and Supplementary Data 4 and 5).

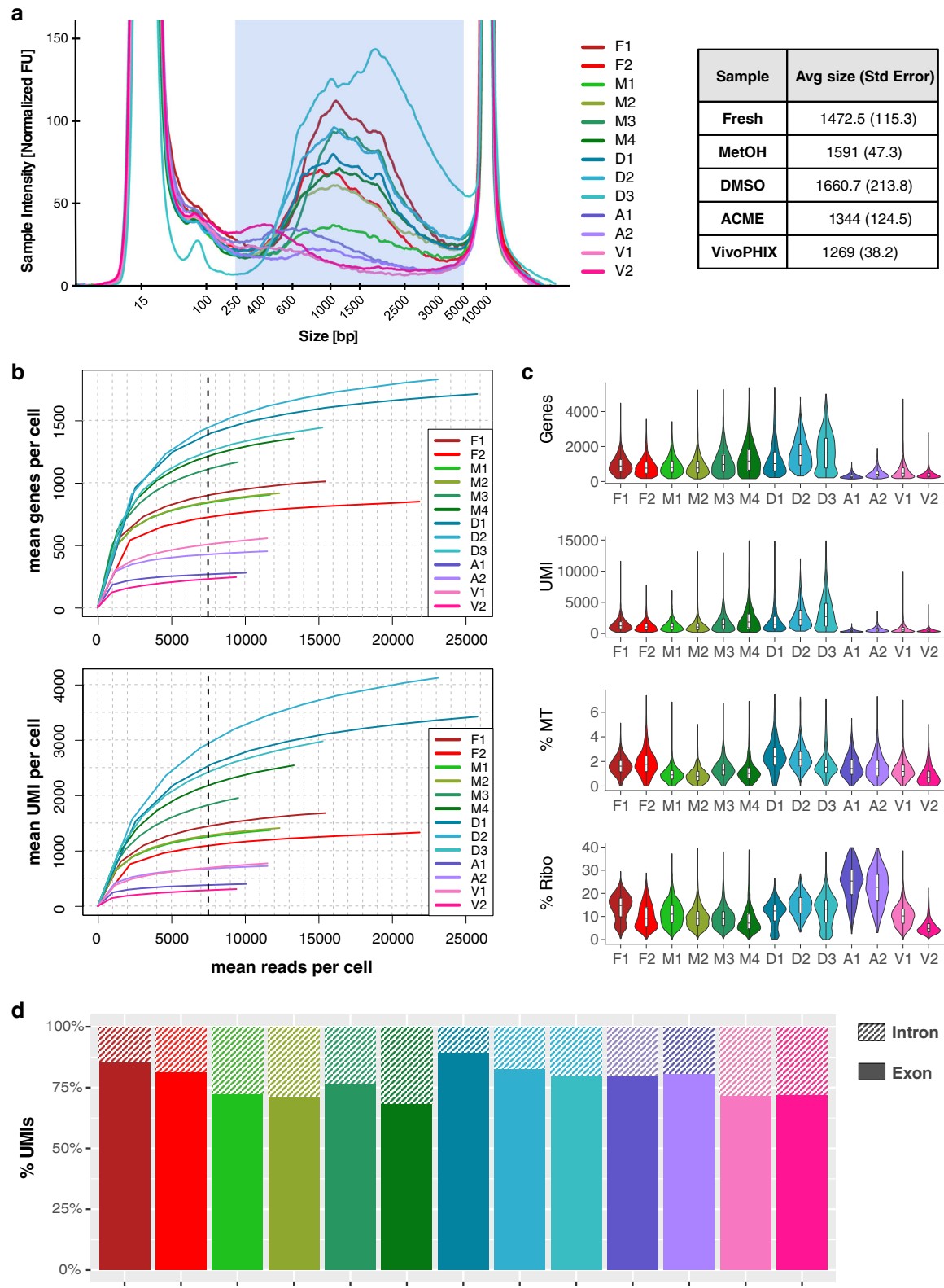

All the clusters identified were present in all the individual samples. Yet, the relative abundance of each of the cell populations changed depending on the preservation method used (Fig. 4a and Supplementary Fig. 6). To investigate if these changes were due to experimental biases or they were systematic biases due to the fixation method, we performed a compositional analysis of the samples with scCODA, a recently developed tool that can reliably identify changes in single-cell datasets even with a low number of replicates[5]. In contrast to other methods, scCODA models compositional bias of the sample as a whole and not for each cluster independently. This prevents wrongly identifying cell proportion changes due to the depletion of a single-cell population, which would artificially result in the increase of all other cell populations in the sample. To identify compositional changes, we

**Fig. 2 Systematic comparison of the effects of cellular preservation on library quality. a** Trace of the libraries obtained after encapsulation, including fresh (F1 and F2), methanol (M1, M2, M3, and M4), DMSO (D1, D2, and D3), ACME (A1 and A2), and vivoPHIX (V1 and V2) samples. The libraries of the samples fixed with vivoPHIX and ACME have smaller fragment sizes and less RNA, which is consistent with RNA degradation. **b** Downsampling plots showing the number of genes and UMIs as a function of sequencing depth (mean reads per cell). In both cases, DMSO (blue) and methanol (green) libraries have a depth equivalent or higher than that of fresh samples (red). **c** Violin plots showing the number of genes, UMIs, the percentage of mitochondrial content (% MT), and ribosomal content (% Ribo) of the cells for each sample after discarding low-quality cells and doublets. The boxplots included inside the violin plots summarize the data distribution. The upper and lower sides of the box represent the 1st and 3rd quartiles. The line in the middle corresponds to the median. Lines extend no further than 1.5 the interquartile range. **d** Barplots showing the percentage of intronic and exonic UMIs assigned to cells for each of the libraries. The number of intronic reads range from ~10% in DMSO sample D1 to ~30% in methanol sample M4. High intronic UMI fractions are indicative of cellular RNA leakage or nuclear RNA enrichment.

**Table 1 Cell-quality statistics.**

| Sample | Cells | Filtered cells | % cells kept | Mean genes per cell | Mean UMI per cell | % MT genes | % Ribo genes |
|---|---|---|---|---|---|---|---|
| F1 | 1764 | 1611 | 91.33 | 1043 | 1738 | 1.65 | 14.1 |
| F2 | 1327 | 1142 | 86.06 | 889 | 1390 | 1.90 | 10.1 |
| M1 | 1841 | 1613 | 87.62 | 970 | 1479 | 0.961 | 11.3 |
| M2 | 1752 | 1568 | 89.50 | 954 | 1469 | 0.886 | 9.60 |
| M3 | 1833 | 1697 | 92.58 | 1185 | 1977 | 1.39 | 9.71 |
| M4 | 1305 | 1172 | 89.81 | 1364 | 2532 | 1.16 | 8.61 |
| D1 | 1234 | 1139 | 92.30 | 1692 | 3271 | 2.31 | 14.8 |
| D2 | 1342 | 1228 | 91.51 | 1787 | 3943 | 1.78 | 12.4 |
| D3 | 1564 | 1405 | 89.83 | 1352 | 2577 | 2.49 | 11.4 |
| A1 | 1313 | 729 | 55.52 | 373 | 545 | 1.61 | 24.5 |
| A2 | 1524 | 1121 | 73.56 | 535 | 866 | 1.55 | 22.4 |
| V1 | 1573 | 1122 | 71.33 | 669 | 937 | 1.31 | 10.6 |
| V2 | 3053 | 1323 | 43.33 | 416 | 529 | 0.877 | 5.51 |

chose as reference group fresh samples, so that the results will indicate if we find compositional changes relative to this group. The results from this analysis highlight a significant depletion of excitatory neurons in DMSO cryopreserved samples compared to fresh samples, while we do not find significant compositional biases in the samples preserved using any of the other methods (Fig. 4b).

**Preservation protocol affects gene expression across cell populations**. Finally, we investigated if preservation methods induced changes in gene expression that could affect the comparison among samples. A comparison of gene expression across samples shows high correlation among all samples (Pearson correlation coefficient $R >= 0.8$), although ACME and vivoPHIX samples have slightly lower correlations with all other samples (Supplementary Fig. 7). This lower correlation can be explained by global or cell-type-specific changes in gene expression but also by compositional biases. To investigate this further, we generated pseudobulk counts for each of the clusters for each sample separately and performed a correlation analysis at the cluster level. Our analyses show that the fixation method induces biases in the clustering of cell populations across samples (Supplementary Fig. 8). However, the high similarity between different clusters, i.e., NPC populations, makes cell clusters preserved with a particular method cluster together. To address this issue, we evaluated the clustering of samples for each cell cluster independently using sigclust2[29], a statistical method designed to test the statistical significance of hierarchical clustering. As can be seen in Fig. 5, in all cases methanol samples clustered with fresh samples. In contrast, in 8 of the 12 cell populations identified, several vivoPHIX and ACME samples cluster separately from fresh samples. This analysis thus confirms that the overall expression profile of methanol-fixed cells in each cluster is more similar to that of fresh cells than that of cells preserved using other methods.

Previous studies have shown that dissociation and preservation methods can induce cellular stress that is reflected at the transcriptomic level[9,30]. Accordingly, we checked the expression of Immediate Early genes (IEGs) and apoptosis markers in our datasets. The apoptosis gene signature was higher in ACME, vivoPHIX, and DMSO samples compared to fresh samples, and higher in DMSO than methanol-fixed samples, although these differences were minimal (Fig. 6a). All fixed samples had higher expression of IEGs than fresh samples, although DMSO cryopreserved cells showed higher expression of IEGs than all the other samples (Fig. 6b). This result indicates that freezing and thawing stresses cells in a way that is globally reflected on the transcriptomic profile of cells. To investigate if cell preservation induced additional expression biases at the individual cluster level, we used muscat[31] to identify cell-type-specific differentially expressed genes (DEGs) in fixed samples compared to fresh (Supplementary Data 6). Our analysis shows that the number of differentially expressed genes (DEGs) is very different across fixation methods. Whereas vivoPHIX samples present many DEGs in all clusters, the amount of significant DEGs is close to zero in DMSO samples (Fig. 6c and Supplementary Data 6). We used gene ontology term enrichment analysis to investigate if different fixation methods would introduce biases in gene expression related to particular functions. Our results did not find any significant terms overrepresented in genes consistently up or downregulated across multiple cell clusters, suggesting that the effect that the fixation methods have on gene expression across clusters is not linked to particular cell functions or locations.

## Discussion

Single-cell transcriptomics methods are becoming the new standard to study transcriptomic changes across samples and conditions. These technologies are relatively new compared to bulk transcriptomics methods such as RNA-seq or 3' seq. Thus, in

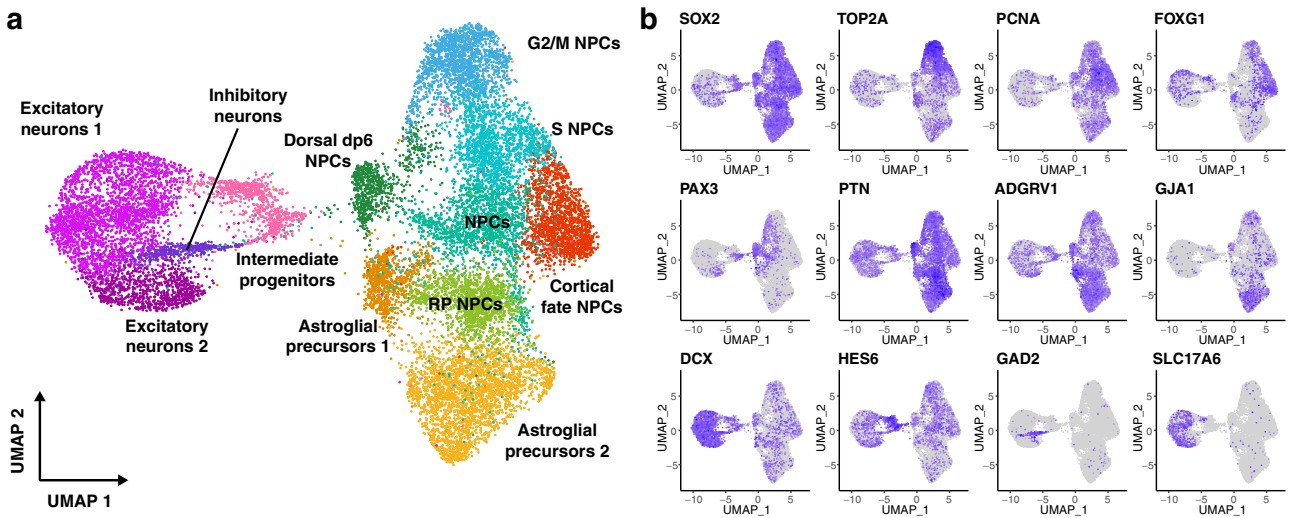

**Fig. 3 scRNA-seq identifies diverse progenitor and immature neuron populations. a** UMAP plot showing the cell populations identified in the NPC samples, which include different types of NPC populations, astroglial precursors, intermediate progenitors, and both excitatory and inhibitory immature neurons. **b** Feature plots of known marker genes that have been used to identify the cell populations in (**a**) including NPC markers (SOX2), proliferation markers (TOP2A and PCNA), cortical fate markers (FOXG1), dorsal fate markers (PAX3), roof plate markers (PTN), astroglial markers (ADGRV1 and GJA1), immature neuronal markers (DCX), intermediate progenitor markers (HES6), inhibitory (GAD2), and excitatory (SLC17A6) neuronal markers.

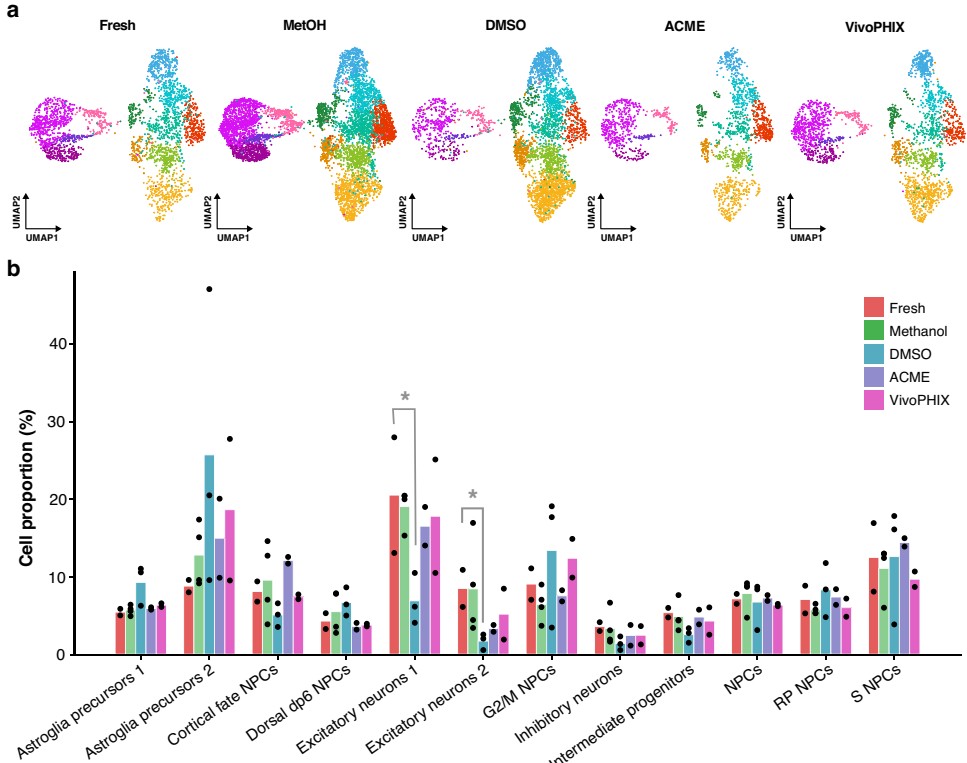

**Fig. 4 DMSO cryopreservation impacts the cell composition of scRNA-seq datasets. a** UMAP plots showing the distribution of cells across clusters for each of the fixation methods. Clusters are colored as in Fig. 3a. DMSO-preserved samples show a strong depletion of cells in neuronal clusters compared to all other samples. **b** Barplot showing the average proportion of cells assigned to each cluster for each method. The height of the bars represents the average for all samples preserved using the same method. Black dots represent the cell proportion in individual samples. * Indicates samples with a significant difference in cell proportion in a particular sample relative to fresh samples as predicted by scCODA, which means that the average score of the sample is below zero according to the scCODA model considering a false discovery rate <0.05.

many cases there are not yet standard preparation protocols for the different samples used. In this project, we have compared how commonly used preservation and fixation methods affect the cell composition and expression of neural and glial cell populations

derived from hiPSCs. This work thus extends previous studies that have compared the effects of only one preservation method on the quality of single-cell transcriptomes or that have focused on their effects on different cell types and thus may not be

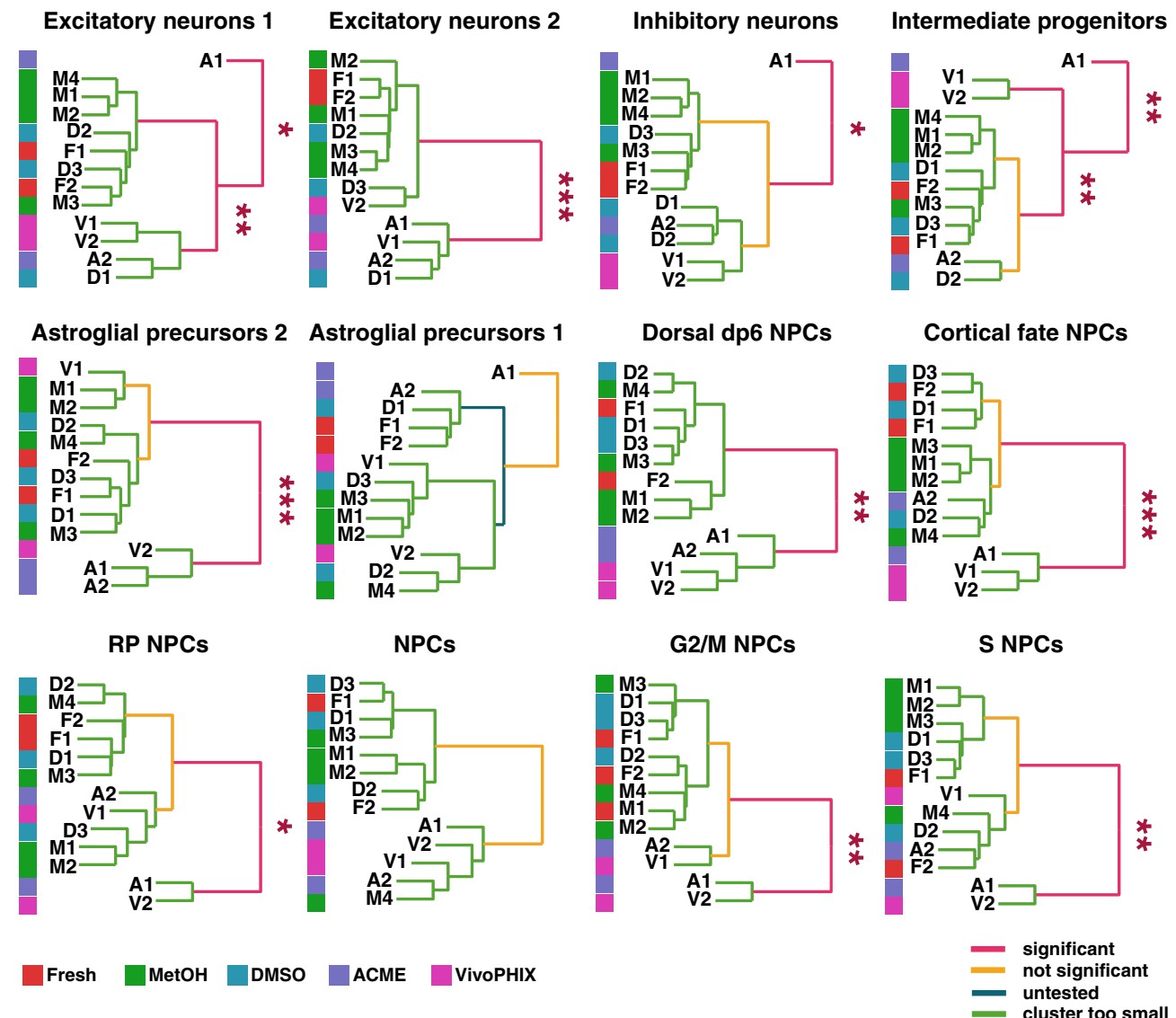

**Fig. 5 Computational analysis identifies fixation-related biases in cell clustering.** Dendrogram showing the similarity of the gene expression profiles of the cells from the different samples belonging to the same cell cluster. The samples are labeled as follows: fresh (F1 and F2, red color), methanol (M1, M2, M3, and M4, green color), DMSO (D1, D2, and D3, blue color), ACME (A1 and A2, purple color) and vivoPHIX (V1 and V2, magenta color). The significance in the hierarchical clustering is assessed using the shc test implemented in sigclust2. Significant branches in the dendrogram are highlighted in pink. *, **, and *** mark the branches with a $P$ value smaller than 0.05, 0.01, or 0.001, respectively. Not-significant branches are highlighted in yellow and not-tested branches in blue and green.

applicable to neural cells[13–19]. Our results show that different preservation/fixation methods affect the quality of the single-cell transcriptomics datasets in different ways, including decreased library complexity, changes in cell composition, and alterations in the expression profile of individual cells (Table 2).

In terms of library complexity, ACME and vivoPHIX samples show a strong decrease in the amount of cDNA obtained after single-cell encapsulation (Fig. 2a), which is also reflected in a lower detection of genes and UMIs (Fig. 2b). In the case of the vivoPHIX samples, this could be related with an initial lower quality of the RNA sample, which had lower RIN values (Supplementary Fig. 1). In the case of ACME samples, which had an RNA quality equivalent to that of fresh samples, this decrease is consistent with previous reports that show that RNA integrity drops in ACME fixed samples over time[10]. The lower library complexity of these samples due to a higher dropout rate is likely the cause of the biases in the cell population clustering

analysis (Fig. 5) and it could contribute to the number of DEGs identified in each cluster (Fig. 6c and Supplementary Data 6).

When looking at cell composition, our results clearly highlight a strong depletion of neuronal cells in the DMSO cryopreserved samples (Fig. 4). Different cell lines and experiments have been used for this project, which could be a confounding effect affecting cell-type composition. However, the comparison of DMSO and Methanol samples that come from the exact same differentiation (M3, M4, D1, and D2) (Supplementary Data 1) highlights a clear difference in the relative abundance of excitatory neuron populations between methanol and DMSO samples, providing additional evidence that the compositional biases are due to fixation/preservation procedure (Supplementary Fig. 6). While DMSO has been previously reported as an excellent method of cell preservation for single-cell transcriptomics[16,17], none of these studies looked at the effect of DMSO on mature

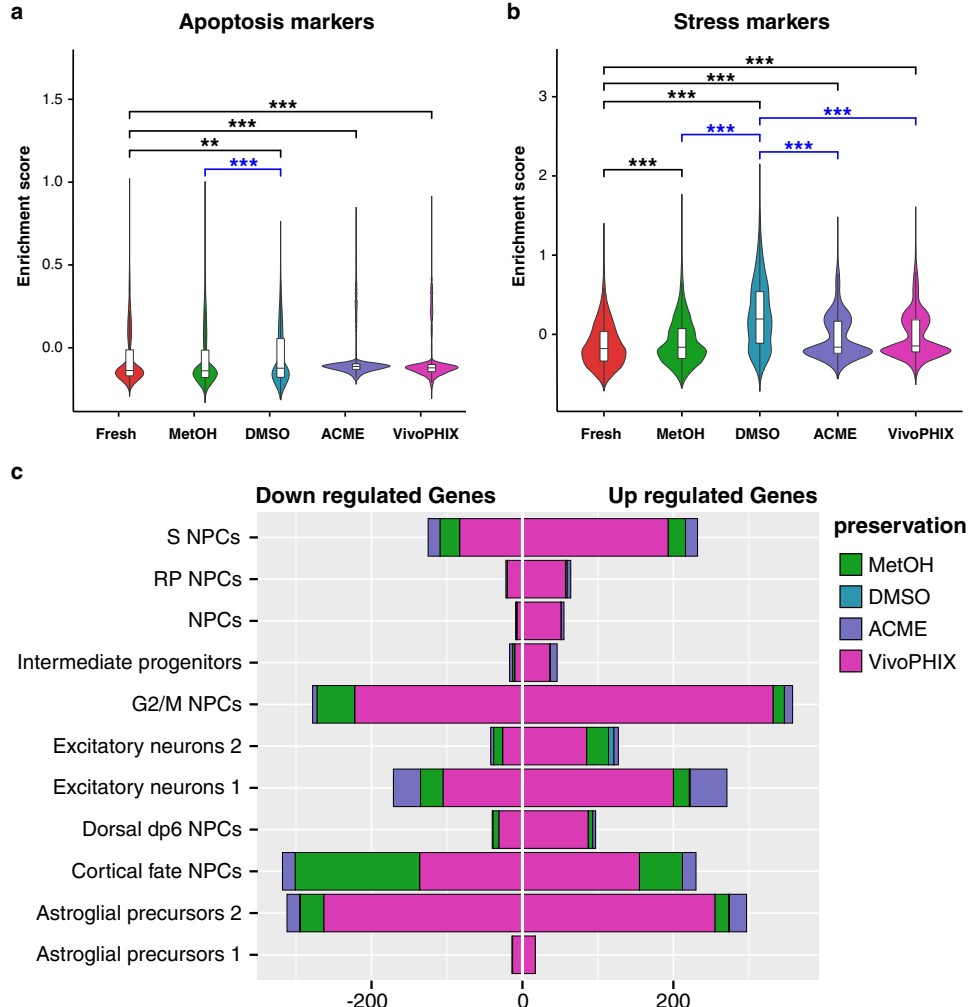

**Fig. 6 Fixation/preservation methods induce sample-specific expression biases. a, b** Violin plots showing the distribution of the enrichment scores of apoptosis (**a**) and stress (**b**) signatures across sample preparation methods. The boxplots included inside the violin plots summarize the data distribution. Upper and lower sides of the box represent the 1st and 3rd quartiles. The line in the middle corresponds to the median. Lines extend no further than 1.5 the interquartile range. **a** The apoptosis enrichment score is higher in DMSO, ACME, and vivoPHIX samples compared to Fresh samples (black asterisks), and higher in DMSO compared to methanol (blue asterisks). **b** The stress signature is higher for all fixation/preservation methods compared to fresh samples (black asterisks), and also higher in DMSO compared to all other fixation methods (blue asterisks). In all cases, statistical significance was tested using a one-tailed Wilcoxon rank-sum test. Comparisons against fresh are marked with black * and comparisons of DMSO against other fixation methods are in blue. ** indicates a *P* value <0.01; *** indicates a *P* value <0.001. **c** Barplot showing the number of significantly upregulated and downregulated genes for each fixation method in each cell cluster with a log2fc >= |0.58| and an adjusted Wald test *P* value <0.05. vivoPHIX and methanol samples have more DEGs across clusters, while the number of DEG in DMSO is close to zero.

**Table 2 Summary of the effects of preservation methods on single-cell libraries.**

| Fixation method | Library yield | Library complexity | Low-quality cells | Composition bias | Stress signature | Apoptosis signature | Expression bias |
|---|---|---|---|---|---|---|---|
| DMSO | HIGH | HIGH | FEW | STRONG | HIGH | MEDIUM | LOW |
| Methanol | HIGH | HIGH | FEW | WEAK | MEDIUM | LOW | LOW |
| ACME | LOW | LOW | MANY | WEAK | MEDIUM | MEDIUM | MEDIUM |
| vivoPHIX | LOW | LOW | MANY | WEAK | MEDIUM | MEDIUM | MEDIUM |

neuronal cells. The reduction in the number of recovered neurons could be due to DMSO toxicity[32,33] or DMSO-induced reactive gliosis, which has been reported previously and can affect cells after a very brief exposure[32], or simply due to the higher fragility of neurons that do not survive thawing/freezing cycles. The latter could explain the higher expression of stress genes in DMSO samples compared to fresh and all other fixed samples. Yet, the obtained neurons do not show strong expression biases as they often cluster with fresh and methanol-fixed samples and have barely no significant DEGs (Fig. 6 and Supplementary Data 6). While our results demonstrate that DMSO is not a good choice for performing compositional analysis of hiPSC-derived cells, it could be used for profiling pure neural samples when cell availability is not an issue.

Our analyses also demonstrate that the different fixation/preservation methods alter gene expression in different ways. Fixation-induced expression biases can affect the overall expression profile of cell clusters, as is the case of vivoPHIX and ACME samples (Fig. 5 and Supplementary Figs. 7 and 8), or drive cell-type-specific changes in gene expression (Fig. 6c and Supplementary Data 6). In addition, we have found a higher expression of IEGs in all samples compared to fresh, and a higher expression of apoptosis markers in vivoPHIX, ACME and DMSO cells compared to fresh (Fig. 6a, b). These results confirm previous observations that associated higher IEG expression to dissociation and cryopreservation biases[9,30], and highlight the need of accurate controls and validation experiments to confirm gene expression changes observed in single-cell data.

Finally, it has to be considered that fixation methods could affect the detection of differentially expressed genes as they affect the library complexity and the number of genes and UMI. Given that the ability to detect differentially expressed genes is directly related to the number of reads or UMIs assigned to a gene, it is possible that subtle differential expression changes due to biological or experimental conditions may be missed. We expect that these effects would be higher in ACME and vivoPHIX samples, which show lower genes and UMIs detected per cell (Fig. 2c), although this can be compensated if more cells are sequenced. Methanol-fixed cells have been successfully used before to discover changes in gene expression in single-cell data across different biological conditions[34,35], indicating that this fixation method is suitable not only to identify homeostatic gene expression but also to identify more subtle biological differences.

In this work, we have tested the impact of fixation and preservation methods in hiPSC-derived neuronal and glial cells using a few replicates (2–4) per condition. This amount of replicates is acceptable considering the cost of individual single-cell experiments and current experimental standards. Yet, it limits our ability to fully assess the impact of all possible variables in the quality of the sample. Our results show that not only the method of preservation affects sample composition and gene expression. Other parameters such as the days of preservation, the cell line used, the differentiation experiment, and the batch of beads have an impact on the final single-cell transcriptomes. Yet, this work does not provide an extensive comparison of all of them, which is out of the scope of this project. Therefore, the results provided in this work may be different when working with different cell types, samples, and single-cell technology or using different experimental conditions that the ones used here. Researchers should consider all these factors and optimize individual experiments given that our results demonstrate that sample processing can impact significantly the results of single-cell transcriptomics experiments.

Taking into account the pros and cons of the different methods (Table 2) and the limitations of this study, our comparative analysis indicates that methanol fixation is the best preservation method to perform single-cell transcriptomics analyses on neural cells. Libraries from methanol-fixed cells have similar complexity to that of fresh cells (Fig. 2) and do no present strong biases in gene expression that affect the overall transcriptomic profile of cells (Figs. 5 and 6 and Supplementary Data 4 and 5) or cell composition (Fig. 4 and Supplementary Fig. 6), thus providing the sample with the most similar profile to that of fresh cells.

## Methods

**hiPSC cell culture and differentiation**. hiPSCs were maintained on 1:40 matrigel (Corning, #354277) coated dishes in supplemented mTeSR-1 medium (StemCell Technologies, #85850) with 500 U ml$^{-1}$ penicillin and 500 mg ml$^{-1}$ streptomycin (Gibco, #15140122). For the differentiation of cortical neurons the protocol described previously[21] was followed with slight modifications. Briefly, hiPSC

colonies were seeded in 12-well plates coated with 1:40 matrigel at a cell density sufficient to ensure 100% confluence one day after plating. At day 1, the medium was switched to neural induction medium (neural maintenance medium (1:1 ratio of DMEM/F-12 GlutaMAX (Gibco, #10565018) and Neurobasal (Gibco, #21103049) medium with 1× N-2 (Gibco, #17502048), 1× B-27 (Gibco, #17504044), 5 µg ml$^{-1}$ insulin (Sigma, #I9278), 1 mM L-glutamine (Gibco, #35050061), 100 µM non-essential amino acids (Lonza, #BE13-114E), 100 µM 2-mercaptoethanol (Gibco, #31350010), 50 U ml$^{-1}$ penicillin and 50 mg ml$^{-1}$ streptomycin) supplemented with 500 ng ml$^{-1}$ noggin (R&D Systems, # 3344-NG-050), 1 µM Dorsomorphin (StemCell technologies, #72102) and 10 µM SB431542 (Calbiochem, # 616461)). The neural induction media was replaced every day for 9–12 days until the neuroepithelial sheet was formed. At this point, the neuroepithelial cells were collected in aggregates using dispase (StemCell Technologies, #07923) and seeded on 20 µg ml$^{-1}$ laminin-coated (Sigma, #L2020) six- well plate containing 2 ml of neural maintenance medium. Cells were incubated in neural maintenance medium with every-other day replacement until neural rosette structures were recognizable (days 12–15 after neural induction). Then, 20 ng ml$^{-1}$ of bFGF (Peprotech, #100-18B) was added to the medium for 2–4 days to promote the expansion of neuro stem cells. At day 18 after neural induction, cells were splitted with dispase for precursors amplification. At day 24, when neurons begin to accumulate at the outside of the rosettes, cells were passaged 1:3 using Accutase (Merck Millipore, #SCR005) in a single-cell suspension and seeded at 50,000 cells cm$^{-2}$ on 20 µg ml$^{-1}$ laminin-coated six-well plate. After a week, cells were split again (ratio 1:4) and seeded on 20 µg ml$^{-1}$ laminin-coated six-well plates and continued the culture for up to 50 days (between 29 and 50) after neural induction with medium changes every second day. Several hiPSCs cell lines and differentiation experiments were used for the obtention of NPCs (Supplementary Data 1). Additional information on the media and reagents used in cell culture can be found in Supplementary Data 7. All the hiPSC cell lines used in this work were generated with informed consent from human donors. The use of hiPSCs in this work was approved by the Spanish' National Commission of guarantees concerning the donation and use of human cells and tissues from the Carlos III National Institute of Health.

**Cell dissociation**. Cells were dissociated into a single-cell suspension following a previously described protocol optimized for scRNA-seq techniques[36]. In summary, cells were enzymatically dissociated for 35 min at 37 °C using papain-accutase dissociation buffer (1:1) (PDS Kit, Papain, Worthington Biochemical Corporation, #LK003176), and quenched with DMEM/F-12 GlutaMAX supplemented with 10 µM of ROCK inhibitor (Y-27632, StemCell Technologies, #72304) and 0.033 mg ml$^{-1}$ of DNase (DNase (D2), Worthington Biochemical Corporation, #LK003170). Additional information on the media and reagents used in cell culture can be found in Supplementary Data 7. Cell suspension was filtered through a 40 µm strainer (Pluriselect Life Science, #43-10040-60) and then centrifuged at 150 g for 3 min at RT. After three washes with 0.4 mg ml$^{-1}$ BSA in DPBS, cells were counted, and viability by trypan blue method was recorded. Only samples with cell viability higher than 75% were included in the study.

**DMSO cryopreserved sample preparation**. Around $2.5 \times 10^6$ cells after dissociation were cryopreserved in cryovials in 1 ml of freezing medium, neural maintenance medium supplemented with 10% v/v of DMSO (Sigma-Aldrich, #D2438) and 20 ng ml$^{-1}$ bFGF. The cryovials were placed into a Mr. Frosty freezing container (Nalgene, #5100-001) previously filled up with isopropyl alcohol and stored at −80 °C overnight (ON) and then transferred for long storage to a vapor phase nitrogen freezer. DMSO cryopreserved samples were thawed in a water bath at 37 °C in continuous agitation, then 1 ml of maintenance medium was added to the vial and transferred to a falcon tube with 10 ml of maintenance medium. Cells were centrifuged at 160 g for 5 min at RT. The supernatant was carefully removed, and the cell pellet was washed with 1 ml of DPBS and 0.01% BSA and then transferred to a 1.5-ml DNA LoBind tube (Eppendorf, #022431021). Cells were pelleted again and resuspended in DPBS and 0.01% BSA. Finally, cells were filtered through a 40 µm strainer and counted in a Neubauer chamber using the standard trypan blue method.

**Methanol sample preparation**. Following the methanol fixation protocol for single-cell RNA-seq in 10X Genomics (CG000136), 200 µl of ice-cold DPBS was added to resuspend a $2.5 \times 10^6$ cell pellet. 800 µl of pre-chilled 100 % methanol was added dropwise until the final methanol concentration reached 80%. Samples in DNA LoBind tubes were placed on ice 30 min, then ON at −20 °C and finally transferred to −80 °C for long storage. Methanol-fixed cells were thawed on ice and centrifuged to remove the supernatant. The cell pellet was washed and rehydrated in 1 ml of DPBS with 0.01% BSA and 0.2 U µl$^{-1}$ of RNase inhibitor (Takara Bio, #2313 A) and 1 mM DTT (Sigma-Aldrich, #D0632)) to avoid RNA degradation. Cells were filtered again with a 40 µm strainer and counted in a Neubauer chamber.

**ACME sample preparation**. A pellet of $1 \times 10^6$ to $5 \times 10^6$ cells was gently resuspended with 100 µl of wash buffer (DPBS with 0.01% BSA and 0.2 U µl$^{-1}$ of RNase inhibitor and 1 mM DTT). Then, ACME solution (wash buffer: methanol: acetic acid: glycerol; in a final ratio of 13:3:2:2) was added dropwise while mixing the tube

to a final volume of 1 ml and incubated for 30 min at RT. After centrifugation at 1000 g at 4 °C for 5 min and discarding the supernatant, the fixed cell pellet was washed twice in 1 ml of wash buffer and resuspended in 1 ml of wash buffer supplemented with 10% v/v DMSO for storage at −80 °C. ACME fixed samples were thawed and rehydrated following the same protocol as methanol-fixed samples.

**vivoPHIX sample preparation**. A cell pellet containing $1 \times 10^6$ to $5 \times 10^6$ cells was gently resuspended with 25 μl of DPBS with 0.01% BSA. For sample fixation, 75 μl of vivoPHIX reagent (Rapid Labs, #RD-VIVO-5) (3:1 ratio) was added and mixed by inverting the tube 10 times during the first 5 min of fixation. Next, the tube was placed on a wheel device at RT and incubated for 30 min more. Samples were stored at 4 °C ON and then transferred to −80 °C for long storage.

vivoPHIX-fixed samples were rehydrated by the addition of one volume (100 μl) of 100% ethanol and mixing the tube several times by inversion. Then, cells were pelleted at 1000 g for 5 min at RT, and the supernatant was discarded. 0.5 ml of vivoPHIX-SCAA (1 volume of vivoPHIX with three volumes of glacial acetic acid) was added very slowly to the cell pellet without disturbing it and incubated for exactly 3 min at RT. The vivoPHIX-SCAA from the pellet was removed and cells were pelleted again at 100 g for 5 min at RT to remove any remaining liquid using a P20 pipette tip. The cell pellet was washed three times with DPBS with 0.01% BSA and 0.2 U μl$^{-1}$ of RNase inhibitor and 1 mM DTT, and the supernatant was discarded. Cells were filtered with a 40 μm strainer and counted in a Neubauer chamber.

**CellCover sample preparation**. A cell pellet containing $1 \times 10^6$ to $5 \times 10^6$ cells was gently resuspended by flicking the tube with the remaining wash buffer (25 μl) after enzymatic dissociation with accutase-papain solution. Then, 10 volumes of Cell-Cover (250 μl) (Anacyte Laboratories) were added and the cell suspension was stored at 4° or −80 °C until use. The recommended protocol provided for the company does not suggest freezing the samples or storing them for a longer period of 2–7 days. However, we wanted to test the efficiency of this reagent in our workflow since the protocol is quite simple. CellCover fixed samples were recovered following the same procedure as for methanol-fixed samples.

**Single-cell capture and library preparation**. For a single-cell encapsulation in a NADIA instrument (Dolomite Bio, #3200590), we followed the protocol provided by the company. We loaded 75.000 cells in a volume of 250 μl (300.000 cells ml$^{-1}$) and 150.000 Macosko oligodT beads (ChemGenes Corporation, #Macosko-2011-10 (V +)) in 250 μl (600 beads μl$^{-1}$) previously washed and resuspended in lysis buffer (6% w/v Ficoll PM-400, 0.2% v/v Sarkosyl, 0.02 M EDTA, 0.2 M Tris pH 7.5 and 0.05 M DTT in nuclease-free water). Cells and beads co-flowed in the microfluidic chip of the device with a capture efficiency of 5–7%.

Immediately after the droplet emulsion breakage, the RNAs captured by the oligodT are reverse transcribed (maxima H RT Master Mix, Thermo, #EP0751) (Supplementary Data 8). Then, the excess bead primers that did not capture an RNA molecule was removed by the incubation of the beads with Exonuclease I (New England Biolabs, #174M0293L) for 45 min at 37 °C. Collected single-cell transcriptomes attached to microparticles (STAMPS) were counted and resuspended in nuclease-free water at 400 beads μl$^{-1}$ and split in pools of 4000 beads per PCR tube and amplified for 9 or 11 PCR cycles depending on the bead batch used for the encapsulation (9 cycles for batch 01, 11 for the others). After cDNA purification with 0.6:1 AMPure XP Beads (Agencourt, #A63881) to sample, quantification with Qubit dsDNA HS Assay (Thermo, #Q32851) and fragment size check-up using a 4200 TapeStation System (Agilent, #G2991BA) was performed. Nextera XT DNA Library Prep Kit (Illumina, #FC-131-1096) was used for the tagmentation of 600 pg of cDNA, illumina adapter tagging and amplification (Supplementary Data 8). The size of Nextera libraries after being purified with 0.6:1 AMPure XP Beads to sample was determined using a 4200 TapeStation System and quantified with Qubit dsDNA HS Assay.1.8 pM of pooled libraries was sequenced on Illumina NextSeq 550 sequencer using Nextseq 550 High Output v2 kit (75 cycles) (Illumina, #20024906) in paired-end mode; 20 bp for Read 1 using the custom primer Read1CustSeqB[37] (cell barcode and UMI) and 64 bp for Read 2, and 8 bp for i7 index.

**scRNA-seq data pre-processing**. scRNA-seq libraries were processed using Drop-seq_tools 2.3 pipeline[38] to generate Digital Gene Expression (DGE) matrices. First, Drop-seq tools were used to generate the index and the annotation files for the hg38 assembly version of the human genome using Ensembl version 100 annotation[39] as reference. Next, fastq files containing paired-end reads were merged into a single unaligned BAM file using picard tools v2.18.14[40]. Using the Drop-seq toolkit with default parameters, reads were then tagged with the cell and the molecular barcodes, trimmed at the 5′ end to remove adapter sequences and at the 3′ end to remove polyA tails. Next, reads were mapped to the human genome (version hg38) with STAR version 2.7.0.a[41]. Resulting bam files were tagged with the annotation metadata files to identify reads overlapping genes. Finally, cell barcode correction was done using the programs DetectBeadSubstitutionError and DetectBeadSynthesisErrors also with default parameters. To estimate the number of cells obtained during the single-cell encapsulation, we used a knee plot using as input the number of uniquely mapped reads assigned to the top N barcodes, where N is at least five times the number of expected cells. The estimated number of cells obtained with this procedure was then used to generate a DGE. Two DGE matrices were generated for each dataset, one containing all UMIs overlapping genes using the parameters LOCUS_FUNCTION_LIST = INTRONIC LOCUS_FUNC-TION_LIST = INTERGENIC and another one containing all UMIs overlapping introns using the parameters LOCUS_FUNCTION_LIST = null LOCUS_FUNCTION_LIST = INTRONIC.

**Filtering of low-quality cells and doublets**. DGE expression matrices were analyzed using Seurat v 4.2.1[42]. First, we generated Seurat objects for each dataset and merged these objects prior to perform the filtering of low-quality cells. After manual inspection, all the cells with a UMI count below 200 or above 17000, a gene number below 200 or above 5500, a percentage of mitochondrial transcripts higher than 7.5%, and a ribosomal content higher than 40% were discarded. The number of cells discarded at each step is provided in Supplementary Data 3. Then, we used DoubletFinder[43] on each sample object separately to remove doublets. The parameters and doublets identified in each dataset are detailed in Supplementary Data 9. After doublet removal, we merged individual objects to perform a joint analysis. Initially, all genes expressed in less than three cells were removed. In addition, we fitted a linear model to describe the relationship between the log number of UMIs and the log number of genes detected per cell. All cells with a residual smaller than −0.5 (3 cells) were discarded. The final Seurat object obtained contained 16,870 cells and 24,468 genes.

**Identification of cell populations**. We used *Seurat* function to regress out the percentage of mitochondrial transcripts, the number of genes, the number of UMIs, and the preservation method. To normalize data, we used the LogNormalize method and multiplied by a scale factor of 10,000. We then selected the 2000 most variable genes to calculate 100 principal components (PCs). We used the ElbowPlot function to manually inspect the amount of variability explained by each PC and select the first 20 PCs that were used to build the kNN graph and compute the UMAP plot using 500 training epochs (iterations). To eliminate the batch effects affecting the identification of shared cell populations across datasets, we used Harmony package[28]. The function RunHarmony was applied on the filtered and processed object, providing the samples as the variable to integrate. By inspecting the updated Elbow plot, we selected the first 19 corrected PCs to perform the clustering. We used the package clustree[44] to inspect the clustering results at different resolutions from 0.1 to 1 and chose a final resolution of resolution 0.7 where we obtained 12-cell populations. To calculate the top markers for each cluster, we used FindAllMarkers function from Seurat with only positive markers and the rest default parameters. Statistically significant markers with an adjusted Wilcoxon rank-sum test P value smaller than 0.05 were selected.

**Stress and apoptosis gene signatures**. We used the function AddModuleScore with default parameters from Seurat package[42] to assess if the different fixation methods induced stress or favored apoptosis among the cells. This function compares the expression of a set of given genes with random sets of genes with similar expression in the dataset to calculate an enrichment. For the apoptosis signature, we built a gene signature including the genes *BCL2, TNF, TP53, CASP3, BAX, CASP8, FAS*[45]. For the stress signature, we used the following immediate early genes *FOS, JUN, EGR1, UBC, HSPA1B, BTG2, IER2, ID3*[30]. Statistically significant differences in the signature score between the different fixation methods and fresh samples was calculated using a one-tailed Wilcoxon rank-sum test.

**Cell composition analysis**. To assess changes in cell composition we used scCODA[5]. To run scCODA we defined fresh samples as a reference condition. To set the comparison, a cluster with low variability across samples had to be chosen as a reference. In this case, we used as reference the NPC cluster, which had a good number of cells and a very low amount of dispersion (expressed as differences between groups). To ensure the results were consistent and reproducible we ran scCODA[5] ten times using the Hamiltonian Monte Carlo sampling method with default parameters and averaged the results. Cell types with average scores below (or above) zero have a significant decrease (or increase) in abundance according to scCODA model (false discovery rate <0.05).

**Differential expression analysis**. We used the function aggregateData from the R package Muscat[31] to obtain pseudobulk expression values for each cluster in each of the samples. Then, we used the function pbDS to perform differential gene expression analysis using DESeq2[46] for each cluster and identify DEGs for each method in comparison to fresh samples. DEGs with an adjusted Wald test P value <0.05 and an absolute log2 Fold change >0.58 are available in Supplementary Data 6 and in Fig. 6c. To assess if the different fixation/preservation methods introduced biases in the expression of particular gene sets, we investigated the biological processes associated to up and downregulated genes using the enrichr function of the GSEApy package[47]. For this analysis, we used all genes that were consistently up- or downregulated across at least four cell types for each fixation/preservation method. Only gene sets with at least ten genes were analyzed. As background set for the enrichment analysis, we provided all genes expressed in the

dataset which had at least 445 UMIs, which is the minimum expression among the DEGs identified. Across all comparisons, we did not identify any significantly overrepresented gene ontology term (hypergeometric test adjusted $P$ value <0.001).

**Correlation analysis**. To assess the correlation across datasets, we compared the expression profiles of all cells from each dataset globally and at the cluster level. For that purpose, we computed pseudobulk expression values for all genes within a specific cluster/dataset. Afterwards, we log transformed the counts $c$ using a pseudocount so that the normalized expression $n$ was $n = log (c + 1)$. We used these normalized expression values to calculate the Pearson correlation coefficient per cell type/sample using cor function in R. We used the pheatmap[48] function to perform a hierarchical clustering using a complete clustering method and using the correlation coefficients to calculate Euclidean distances between clusters. Given that many cell clusters are very similar, we repeated the same procedure for each cell type separately and determined the statistical significance of sample clusters using the shc function from the sigclust2 R package[29] on the corresponding correlation table. The shc method uses a Monte Carlo simulation-based significance testing procedure to assess the significance of the hierarchical clustering results of the dataset. Statistical significance is evaluated at each node along the hierarchical tree (dendrogram) starting from the root using a gaussian null hypothesis test, and a corresponding $P$ value is calculated using the 2-means cluster index, a statistic sensitive to the null and alternative hypotheses. A family-wise error rate controlling the procedure is applied to correct for multiple testing. We generated dendrogram plots for each cell type to illustrate the similarity among clusters for different samples and highlight the statistically significant differences.

**Measuring the capture efficiency of bead batches**. Different batches of beads can have different mRNA capture efficiency. In order to evaluate the impact of using different batches in scRNA-seq encapsulations, we measured the capture efficiency of bead batches 01 and 02. We encapsulated the same sample twice using both bead batches used in the paper. After RT-PCR, we amplified 4000 STAMPS by PCR using 9, 10, 11, or 12 cycles independently. After AMPure XP Beads purification, the cDNA of each PCR was quantified by Qubit dsDNA HS Assay. Our results demonstrate that to obtain a similar cDNA concentration with the two bead batches, we needed to increase by 2 the number of PCR cycles when using batch 02 (Supplementary Fig. 2).

**Statistics and reproducibility**. All single-cell transcriptomics experiments have been performed using at least two different cell lines and two independent differentiation experiments. Samples D1, D2, M3, and M4 come from the same differentiation experiments but were fixed using different protocols and at different days (DMSO cryopreservation for D1 and D2, Methanol fixation for M3 and M4). F1, F2, M1, M2, D3, A1, A2, V1, and V2 samples all come from independent differentiation experiments. All details about days of differentiation, cell lines used, and other details from the sample preservation can be found in Supplementary Data 1. The initial number of cells of each of the samples and the final number after quality filtering is provided in Table 1. These cells are the ones used in all the analyses provided in the article.

Most computational analyses have been performed using R[49] and specific packages implemented in R 4.2.1 such as Seurat[50], Harmony[28], clustree[44], muscat[31], sigclust2[29], and DoubletFinder[43]. We have also used the program scCODA[5] implemented in Python to assess changes in cell-type abundances and the program GSEApy[47] to perform a GO-term enrichment analysis. All the details are provided in the Methods section.

**Reporting summary**. Further information on research design is available in the Nature Portfolio Reporting Summary linked to this article.

## Data availability
All raw and processed scRNA-seq data generated for this study can be found in GEO database under accession number GSE209947. Source data underlying Figs. 2c, d, 4b, and 6 are available in the Supplementary Data files 10–14.

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

## Acknowledgements

We thank all members from the Plass Lab for useful comments and critical discussion. We also thank Yvonne Richaud-Patin from the Regenerative Medicine Program at IDIBELL and Dr. Zomeño from IDIBELL's Advanced Cell and Tissue Culture platform for their help with iPSC cell culture. We thank Dr. Iglesias for critical review of the manuscript. We also thank Dr. Iglesias, Dr. Kim and Dr. Sebé-Pedrós for their support in setting up ACME and vivoPHIX protocols for cell preservation. This research was funded by a research project from the State R&D Program Research Challenges from the Spanish Ministry of Science, Innovation and Universities (Grant number: PID2019-108580RA-I00/AEI/10.13039/501100011033). M.P. work is supported by a Ramón y Cajal contract of the Spanish Ministry of Science and Innovation (RYC2018-024564-I). We thank CERCA Program/Generalitat de Catalunya for IDIBELL institutional support.

## Author contributions

A.G.F. and M.P. designed the project and planned experiments. A.G.F. and N.C. performed experimental work. F.A., M.H., and M.P. performed computational data analyses. L.M. performed the compositional analysis of the samples. M.P. acquired funding, supervised, and coordinated the work. A.G.F., F.A., and M.P. interpreted the results. M.P. wrote the manuscript with input from all authors.

## Competing interests

M.P. is an Editorial Board Member for *Communications Biology*, but was not involved in the editorial review of, nor the decision to publish this article. The remaining authors declare no competing interests.
