## [Peer Review File · Communications Biology]

Reviewers' comments:

Reviewer #1 (Remarks to the Author):

This work by Gutiérrez-Franco et al is incredibly practical and timely in a field that is increasingly using single cell or nuclei approaches to characterize a diverse range of biological systems. Thus, these results are timely and helpful, but a few points of clarity should be addressed prior to publication.

1) The labels used throughout the manuscript are hard to decipher. Having keys within each figure and in the legend for all abbreviations, labeling all axes, and keep clear color schemes through the paper will be helpful.

2) The numbers for QC metrics like genes/cell or % mito are shown, but indicating what fraction of each dataset is lost to these metrics usually common QC cutoffs would be very helpful.

3) For the cell compositions, using stacked barplots might make it easier to note differences?

4) Figure 5 is hard to see and interpret, could be clarified or excluded.

5) Differential gene expression analysis between methods and pathway analysis of the changes could be helpful for people who want to regress out some of these sources of variation if there are consistent artifact signatures.

6) Including much much clearer protocols for each method used with all necessary details (volumes, brands, concentrations, times, caveats) would be helpful so others can reproduce the methods that best fit their experiment.

Reviewer #2 (Remarks to the Author):

In this manuscript, Gutiérrez-Franco et al. perform a timely analysis of the effect of fixation/preservation methods for scRNA-seq of neural cells. The core idea is to avoid any aberrant gene expression (e.g. induction of IEGs), shift in cell type composition or changes in the quality score leading to cell drop out during QC (e.g. a shift in the expression of mt-genes or RPL genes) as a consequence of cell handling prior to library prep.

Following enzymatic digestion, they compared freshly dissociated cells to cells fixed with a range of different fixatives. They then perform sample QC, basic clustering and correlation analysis across all conditions to identify the optimal processing conditions for neural cells. They argue that across all parameters assessed, methanol fixation performed best for neural cells.

There is currently no gold standard for sample processing, preservation and QC pipelines in scRNA-seq, and benchmarking as done in this paper is an important contribution to the ever-expanding field. There are some striking findings in terms of general cell viability and library prep following the different fixations, but the reviewers and readers need to be convinced of both the merit/need of post-dissociation fixation in general, as well as the fixation-induced changes in QC and the presence of bona fide fixation- or dissociation-induced differentially expressed genes (DEGs) that inform the choice of fixative made in this MS.

1 - The premise of this MS is that fixation post-dissociation is critical in avoiding handling effects pre-library prep, however, there should be more discussion on why this is necessary at all. Aberrant

changes in gene expression and a drop in QC are most likely primarily occurring during the enzymatic and physical cell dissociation (as shown for microglia here for example: PMID:35260865). The time following cell dissociation and cell lysis in GEMs (or other droplets) will be minimal assuming there is local processing infrastructure available. Maybe the authors can elaborate on why post-dissociation fixation needs to be considered. For example, post-dissociation fixation could be performed to preserve the cells for processing at later time points (the authors mention e.g. FACS to purify sub-populations before sequencing; or one could be accumulating samples for batch processing) or if the microfluidics is done in a different physical location (i.e. samples are preserved to be shipped to a sequencing/processing core). Maybe the authors can clarify and emphasize in the main text exactly what the fixation is meant to preserve. For example, with a dissociation at 37C for 35 min, as done in this MS, induction of IEGs, which take minutes to induce, won't be rescued by post-dissociation fixation. Indeed, based on Fig5, fixation does not prevent putative dissociation induced-IEG induction or putative changes in apoptosis pathways (although a true ground truth and stats are missing, see points below).

2 - If the authors argue that the point of post-dissociation fixation is to rescue persistent dissociation-induced artefacts (and I am not sure they are), then the ground truth should be non-dissociated cells. Is there any available 'bulk'-seq data of non-processed cultures that could give us an idea whether the fixation helps alleviate any potential aberrant gene expression? A comparison could be made to the pseudobulk data presented in Fig6.

3 - Another unresolved question in the field is how power calculations are done in scRNA-seq. In general, the number of biological replicates is used rather than the number of cells sequenced. Here, the authors have conditions ranging from an n=2 to n=4 (which is understandable and acceptable given the price point of scRNA-seq). However, I think it is important to discuss this limitation in addition to my point (7) below.

Nonetheless, it is unclear how the statement 'DMSO cryopreservation affects the overall expression of stress markers in scRNA-seq data' in the Fig5 legend was tested statistically? One would have to compare the module scores for each sample and test it across the conditions. Given this is one of the major findings of the paper, some form of statistical test has to be applied to convince the readership. Fig 5 also needs a scale to understand the dynamic range of the stress/apoptosis scores.

In general, however, it would be more powerful to calculate DEGs unbiasedly to identify how/whether the different fixation methods affect gene expression (see point (6)).

4 - Fig4: Can the authors extend their explanation how scCONDA identified a shift in cell type composition? This is particularly in reference to restrictive numbers of repeats.

5 - Fig6: The authors argue that ACME and vivoPHIX cluster together but there is no statistical evidence that this is a significant or meaningful observation. In fact, all samples occupy the same area on a 2D space (e.g. Fig4) and can be readily integrated using Harmony, so it's not clear whether this would affect any downstream analysis at all. Additionally, an MDS or PCA plot of all pseudobulked samples would be more intuitive to read than the heatmap in Fig6.

6 - The authors make the valid point that both dissociation and fixation could differentially affect subtypes of cells. It would be powerful to show per cluster (i.e. per cell type) module scores or DEGs to identify potential differences in how sample processing or fixation affect different cell types. Tools like muscat (Bioconductor) can facilitate multi-sample, multi-condition, cluster-resolved differential gene expression using pseudobulk expression levels, which would address most of my points.

7 - There is extensive variability in how long the samples were preserved for before sequencing. It

would be important to show whether the time preserved correlates with the QC readouts used in this MS – i.e. what changes in QC are due to the fixation vs the time the cells spent frozen down.

8 – Another limitation of this study is that it is not assessing whether different fixation methods affect DEG analysis across different biological conditions (e.g. different culture/maturation conditions or drug treatments). While meOH might preserve homeostatic gene expression, it is unclear whether it can recover state-dependent DEGs. This could be discussed in the context of other limitations mentioned above.

Reviewer #3 (Remarks to the Author):

In this manuscript, the authors compare different fixation methods of iPSC-derived neural precursors to determine which strategy is optimal for future scRNA-seq experiments on a droplet-based platform. A detailed comparison is a welcome addition to the field, as the authors correctly note that distinct fixation protocols can have significant differences on the preservation and quality of RNA integrity and scRNA-seq results. Notably, neurons present an additional challenge as they can be more sensitive to fixation, freezing and thawing methods, with many reagents on the market dedicated to preserving neural precursors and mature neurons. While the authors do note some important and striking differences between the different cell preservation strategies, their lack of detail in their methodology complicates my interpretation of some of their findings and makes it difficult to determine how much of the variability is due to the specific fixation reagents vs. other factors the authors may have overlooked. My detailed comments are below:

1. In an ideal scenario, the authors would have generated a large batch of iPSC-derived neurons and then performed multiple fixation protocols on this same batch of cells (possibly being done 1-2 days apart if necessary to minimize complications of performing different freezing experiments on the same day). This strategy would minimize/eliminate any batch-to-batch variability because different freezing conditions would be done on the exact same cultures and thus any differences would almost certainly be due to freezing/thawing preparations. Is this how the authors performed their experiments? Or instead, were all of the authors samples (F1, F2, M1, etc.) performed on separate differentiations? If each freezing experiment was a distinct differentiation, then I'm concerned about how inherent batch-to-batch variability between distinct differentiations could affect their conclusions. Please expand on the methodology used, and if each experiment was a distinct differentiation, then the authors must address this batch-to-batch variability and clearly describe how this affects interpretation of their results.

For example, M3 and D1 (and M4 and D2) have identical differentiation days (40 or 50, respectively), preservation days (60) and cell lines used (AD5), all from Table S1. Were M3 and D1 from identical differentiations, and similarly were M4 and D2 from the same differentiation? If so, the authors should clearly state this, as it makes the differences between DMSO and Methanol conditions clearly dependent on the fixation conditions and not due to batch-to-batch variability.

2. Another important methodological distinction: RNase inhibitor and DTT was included in the Methanol, ACME and vivoPhix preparations, but not in the DMSO preparation. This seems like a pretty important point and a potential (I'd argue likely) causative reason for differences between DMSO and Methanol preps. Do the authors have a specific reason for excluding RNase inhibitor and DTT from the DMSO fixation (i.e., do they not dissolve in the DMSO fixative)? Do the authors think this could contribute to the significant loss of postmitotic neurons in the DMSO fixation procedure? I would really like to see a sample where RNase and DTT are included in the DMSO fixation to have a better idea

how important these reagents are in fixation process.

3. Another confusing thing about the methods description was the variability in 'days of differentiation' and 'days preserved' between the samples. How different is the cell composition (and gene expression) in samples harvested at differentiation day 29 (V1) compared to differentiation day 50 (M4&D2)? Couldn't the difference in differentiation days partially explain for some differences in cell types detected? The authors should discuss this issue in more detail. Similarly, do the authors believe that there are any differences between samples preserved for 7-9 days vs. 120 days? They could generate some insights into this by directly comparing A1 (120 days) vs. A2 (7 days), although it looks like these two samples were generated with different cell lines.

4. The authors appear to use 2 iPS lines (Ctrl and AD), but they don't mention these or describe them in the Methods. Are there differences between these 2 cell lines, do they generate the same proportion of progenitors and neurons, etc. Several sample preparations use only 'AD' iPSCs (DMSO and CellCover) whereas other conditions use both 'Ctrl' and 'AD' iPSCs (Fresh, Methanol, ACME, VivoPHIX). Combining these different conditions complicates interpretation of their data. It might be better to focus solely on AD5 cells since they were used for all sample preparations. For example, I would like to see comparative analysis specifically between M3, M4, D1 and D2, as they all use the same cell line and other factors. Adding M1 and M2 to the Methanol analysis, which were frozen for only 7 days, complicates a direct comparison between the Methanol and DMSO conditions.

5. In Fig S3, the cells fixed with DMSO show distinct, almost non-overlapping profiles compared to the other 4 conditions. I'm not sure why these dot plots are a different shape from the ones in Figure 3-5, but it would be nice if they were from the same plot so that I could know the different cell types (progenitors vs. postmitotic cells). Do the authors know why the DMSO cells are completely separated from the other conditions on this UMAP plot while they are overlap much better in Figures 3-5? Was anything computationally different between processing the data in Fig S3 vs. Figs 3-5? And the authors should reference Fig S3 when discussing that DMSO is known to be toxic to the cells and thus could be causing significant transcriptional changes in the cells.

Reviewers' comments:

Reviewer #1 (Remarks to the Author):

Expertise: scRNA-seq in neurons, neuronal iPSC models

This work by Gutiérrez-Franco et al is **incredibly practical and timely** in a field that is increasingly using single cell or nuclei approaches to characterize a diverse range of biological systems. Thus, these results are timely and helpful, but a few points of clarity should be addressed prior to publication.

1) The labels used throughout the manuscript are hard to decipher. Having keys within each figure and in the legend for all abbreviations, labeling all axes, and keep clear color schemes through the paper will be helpful.

Following the reviewer's suggestion, we have improved the labelling and info in figures and figure captions thorough the manuscript.

2) The numbers for QC metrics like genes/cell or % mito are shown, but indicating what fraction of each dataset is lost to these metrics usually common QC cutoffs would be very helpful.

We have included an additional Table (New Table S3) showing how many cells are discarded due to each individual filtering parameter.

3) For the cell compositions, using stacked barplots might make it easier to note differences?

We have tried doing the stacked barplots. If we use stack plots per sample we do not see the variability within each method (Figure R1) and if we do it for each sample it is harder to see the differences between fixation methods (Figure R2) as it highlights too much the variation across individual experiments (see below). We have decided to keep the barplot as before but also include a new supplementary figure (Fig S6) including the stacked barplots for each sample individually.

Figure R1. Stacked barplot showing the proportion of each cell type in each fixation method. The data show the aggregate distribution for all the samples processed with each experimental protocol and therefore does not reflect the within-group variability.

Figure R2. Stacked barplot showing the proportion of each cell type in each sample. Names of the samples are as follows: Fresh1 (F1), Fresh2 (F2), Methanol 1 (M1), Methanol 2 (M2), Methanol 3 (M3), Methanol 4 (M4), DMSO 1 (D1), DMSO 2 (D2), DMSO 3 (D3), ACME 1 (A1), ACME2 (A2), VivoPHIX 1 (V1) and VivoPHIX 2 (V2). The variability among samples is partially because the experiments include different cell lines (see Table 1) and different biological replicates.

4) Figure 5 is hard to see and interpret, could be clarified or excluded.

We agree with the reviewer that Figure 5 was not clear. In the new Figure 6 (panels a and b), which contains the information in the old Figure 5, we have removed the feature plots and provided the p-values of the comparisons between all samples against fresh (black *) and DMSO against the other fixation methods (blue *). Only comparisons with a significant p-value < 0.05 are shown. The analyses clearly shows that DMSO, ACME and VivoPHIX, but not Methanol, have a higher expression of apoptosis markers compared to Fresh samples. The comparison of stress markers show also a higher expression in all fixed and preserved samples compared to Fresh. Yet, DMSO samples have higher enrichment values than all other samples. We have also modified Table 2, which provided a summary of effects of each of the fixation/preservation methods, to clearly reflect these changes.

New Figure 6 caption: Fixation/preservation methods induce sample-specific expression biases. **a,b)** Violin plots showing the distribution of the enrichment scores of apoptosis (**a**) and stress (**b**) signatures across sample preparation methods. **a)** The apoptosis enrichment score is higher in DMSO, ACME and vivoPHIX samples compared to Fresh samples (black asterisks), and higher in DMSO compared to methanol (blue asterisks). **b)** The stress signature is higher for all fixation/preservation methods compared to fresh samples (black asterisks), and also higher in DMSO compared to all other fixation methods (blue asterisks). In all cases, statistical significance was tested using a one tailed Wilcoxon signed rank test. Comparisons against fresh are marked with black * and comparisons of DMSO against other fixation methods in blue. ** indicates a p-value < 0.01; *** indicates a p-value < 0.001. **c)** Barplot showing the number of significantly upregulated and downregulated genes for each fixation method in each cell cluster with a $\log_2fc \geq |0.58|$ and an adjusted p-value < 0.05. vivoPHIX and methanol samples have more DEGs across clusters while the number of DEG in DMSO is close to zero.

5) Differential gene expression analysis between methods and pathway analysis of the changes could be helpful for people who want to regress out some of these sources of variation if there are consistent artifact signatures.

Following the suggestion of the reviewers 1 and 2, we have now performed a differential gene expression analysis for each cluster individually (Figure 6c, see above, and Table S6). Additionally, for all those genes that were consistently upregulated or downregulated for a specific fixation method across cell types (at least consistently up or down in 4 cell types compared to Fresh samples), we have performed a GO-term enrichment analysis. This analysis highlights that the biases in gene expression introduced by DMSO and ACME are very small while they are stronger for Methanol and specially VivoPHIX fixed samples (Figure 6c). The GO-term enrichment analysis does not show significantly enriched terms in the DEG consistently UP or DOWN regulated across cell types. We have clarified this now in the manuscript.

6) Including much much clearer protocols for each method used with all necessary details (volumes, brands, concentrations, times, caveats) would be helpful so others can reproduce the methods that best fit their experiment.

We have provided a more detailed explanation of the experimental protocols in the methods section, which includes the brands and catalog number of all reagents used. Furthermore, we provide additional supplementary tables (Table S8 and Table S9) including the detailed volumes, times and conditions of the reagents used for cell culture and dissociation (Table S8) and for the library preparation (Table S9).

Reviewer #2 (Remarks to the Author):
Neuronal single-cell genomics

In this manuscript, Gutiérrez-Franco et al. perform a timely analysis of the effect of fixation/preservation methods for scRNA-seq of neural cells. The core idea is to avoid any aberrant gene expression (e.g. induction of IEGs), shift in cell type composition or changes in the quality score leading to cell drop out during QC (e.g. a shift in the expression of mt-genes or RPL genes) as a consequence of cell handling prior to library prep.

Following enzymatic digestion, they compared freshly dissociated cells to cells fixed with a range of different fixatives. They then perform sample QC, basic clustering and correlation analysis across all conditions to identify the optimal processing conditions for neural cells. They argue that across all parameters assessed, methanol fixation performed best for neural cells.

There is currently no gold standard for sample processing, preservation and QC pipelines in scRNA-seq, and benchmarking as done in this paper is an important contribution to the ever-expanding field. There are some striking findings in terms of general cell viability and library prep following the different fixations, but the reviewers and readers need to be convinced of both the merit/need of post-dissociation fixation in general, as well as the fixation-induced changes in QC and the presence of bona fide fixation- or dissociation-induced differentially expressed genes (DEGs) that inform the choice of fixative made in this MS.

1 - The premise of this MS is that fixation post-dissociation is critical in avoiding handling effects pre- library prep, however, there should be more discussion on why this is necessary at all. Aberrant changes in gene expression and a drop in QC are most likely primarily occurring during the enzymatic and physical cell dissociation (as shown for microglia here for example: PMID:35260865). The time following cell dissociation and cell lysis in GEMs (or other droplets) will be minimal assuming there is local processing infrastructure available. Maybe the authors can elaborate on why post-dissociation fixation needs to be considered. For example, post-dissociation fixation could be performed to preserve the cells for processing at later time points (the authors mention e.g. FACS to purify sub-populations before sequencing; or one could be accumulating samples for batch processing) or if the microfluidics is done in a different physical location (i.e. samples are preserved to be shipped to a sequencing/processing core). Maybe the authors can clarify and emphasis in the main text exactly what the fixation is meant to preserve. For example, with a dissociation at 37C for 35 min, as done in this MS, induction of IEGs, which take minutes to induce, won't be rescued by post-dissociation fixation. Indeed, based on Fig5, fixation does not prevent putative dissociation induced-IEG induction or putative changes in apoptosis pathways (although a true ground truth and stats are missing, see points below).

We agree with the reviewer considerations that fixation/preservation methods will not rescue any gene expression changes produced by dissociation methods or previous steps involved in sample handling. As the reviewer points, the idea of fixation/preservation of cells is more related to the preservation of the RNA molecules inside the cell to decouple sample obtention/processing with encapsulation and library preparation. This step is essential to preserve the cells for processing at later time points or to perform batch processing of samples. Following the reviewer's suggestion, we have included a more detailed description of the reasons why fixation could be relevant in the introduction (see below the text added in italics).

As a result, the preparation of samples for scRNA-seq can take several hours, which makes more convenient to process them at later time points. Apart from technical difficulties, cell preservation is also important if we need to decouple sample dissociation and processing for other reasons such as the shipment of samples to an external facility, or if we want to collect multiple samples and process them together at a later time point to save time or money. In all these cases, researchers would like to preserve these samples in a way that minimizes the differences in cell composition and gene expression of individual cells in comparison to the original sample. That is, the best preservation method will be the one that has the smallest impact in the cell composition of the sample and the transcriptomic profile of the individual cells.

2 - If the authors argue that the point of post-dissociation fixation is to rescue persistent dissociation-induced artefacts (and I am not sure they are), then the ground truth should be non-dissociated cells. Is there any available 'bulk'-seq data of non-processed cultures that could give us an idea whether the fixation helps alleviate any potential aberrant gene expression? A comparison could be made to the pseudobulk data presented in Fig6.

As the reviewer pointed, we do not believe that post-dissociation fixation can rescue dissociation-induced artefacts. Dissociation induced artifacts exist and have been previously assessed in other publications (Denisenko et al. 2020; Uniken Venema et al 2022). Instead, the main goal of this manuscript is to discuss exclusively fixation/preservation-induced artifacts. Taking this into account, we believe that the

approach used, i.e. scRNA-seq of single-cell suspensions of cell populations obtained using the same experimental protocol, both for cell culture and for enzymatic dissociation, is the appropriate to assess this.

3 – Another unresolved question in the field is how power calculations are done in scRNA-seq. In general, the number of biological replicates is used rather than the number of cells sequenced. Here, the authors have conditions ranging from an n=2 to n=4 (which is understandable and acceptable given the price point of scRNA-seq). However, I think it is important to discuss this limitation in addition to my point (7) below.

We understand the concerns risen by the reviewers and agree that the limitations of the study has to be discussed. We have added a section named *limitations of this study* in which we discuss these issues.

Limitations of this Study

In this work we have tested the impact of fixation and preservation methods in human iPSC-derived neural and glial cells using a few replicates (2 to 4) per condition. This amount of replicates is acceptable considering the cost of individual single-cell experiments and current experimental standards. Yet, it limits our ability to fully assess the impact of all possible variables in the quality of the sample. Our results show that not only the method of preservation affects sample composition and gene expression. Other parameters such as the days of preservation, the cell line used, the differentiation experiment, and the batch of beads have an impact on the final single-cell transcriptomes. Yet, this work does not provide an extensive comparison of all of them, which is out of the scope of this project. Therefore, the results provided in this work may be different when working with different cell types, samples, single-cell technology or using different experimental conditions that the ones used here. Researchers should consider all these factors and optimize individual experiments given that our results demonstrate that sample processing can impact significantly the results of single-cell transcriptomics experiments.

Nonetheless, it is unclear how the statement ‘DMSO cryopreservation affects the overall expression of stress markers in scRNA-seq data’ in the Fig5 legend was tested statistically? One would have to compare the module scores for each sample and test it across the conditions. Given this is one of the major findings of the paper, some form of statistical test has to be applied to convince the readership. Fig 5 also needs a scale to understand the dynamic range of the stress/apoptosis scores.

We agree with the reviewer that the data shown was not clear. We have now modified Figure 5 (new Figure 6, see above) so that the feature plots have been removed and the violin plots contain a boxplot and a clear label. To assess if the apoptosis or stress signature was higher in the preserved/fixed samples compared to fresh, we have compared the distribution of the enrichment scores obtained with the AddModuleScore function for each of the preservation methods using a one-tailed Wilcoxon signed-rank test. Our results show that DMSO, ACME and VivoPHIX samples have significantly higher apoptosis signature enrichment scores compared to fresh samples (black asterisks), and that DMSO have even higher enrichment compared to methanol samples (blue asterisks). We have done the same test to assess the difference in the enrichment of the stress signature. In this case, all samples have a significant higher enrichment of stress markers compared to Fresh sample, but DMSO have significantly higher values than all

other fixation methods. We have modified the text in the results section and methods section accordingly to clarify this.

In general, however, it would be more powerful to calculate DEGs unbiasedly to identify how/whether the different fixation methods affect gene expression (see point (6)).

We agree with the reviewer. We have followed the reviewer's suggestion of using Muscat to identify per-cluster differentially expressed genes. The results are included in the new Figure 6c and Table S6). Additionally, for all those genes that were consistently upregulated or downregulated for a specific fixation method across cell types (at least consistently up or down in 4 cell types compared to Fresh samples), we have performed a GO-term enrichment analysis, although no significant GO-terms were identified, suggesting that fixation/preservation methods induce cell-type specific gene expression changes or random gene expression biases. This analysis highlights that the biases in gene expression introduced by DMSO and ACME are very small while they are stronger in Methanol and specially VivoPHIX fixed samples. We have included these results in the manuscript. The whole section discussing how fixation methods affect gene expression and sample composition has been rewritten to improve clarity and include the new analysis.

4 – Fig4: Can the authors extend their explanation how scCONDA identified a shift in cell type composition? This is particularly in reference to restrictive numbers of repeats.

scCODA (Büttner et al. 2021) is a computational method that uses a Bayesian model to model the composition of a sample and assess the differences across conditions considering different experimental and methodological limitations of single-cell experiments, particularly the low number of experimental replicates. Although I cannot comment on the mathematical model, the benchmarking of scCODA against other methods that have been developed to assess differential cell abundance clearly shows a much higher Matthews' correlation coefficient (MCC) on synthetic datasets than any other methods, especially when having low number of replicates per group ($n < 3$) (see Figure 2 from that paper included below).

Fig. 2 Comparison of scCODA's benchmark performance to other differential abundance testing methods. Bayesian models (red), non-standard compositional models (blue), compositional tests/regression (green), non-compositional methods (purple). Shaded areas represent 95% confidence intervals. **a** Receiver-operating curve ($n > 1$ samples per group). AUC scores are reported in (Supplementary Table 1). **b** Precision-recall curve ($n > 1$ samples per group). Average precision scores are reported in (Supplementary Table 1). **c-e** Performance metrics with increasing number of replicates per group over all tested scenarios. In the case of $n = 1$ sample per group, only Bayesian methods are applicable, other methods cannot detect any changes. **c** Overall performance measured by Matthews' correlation coefficient (MCC). **d** Sensitivity measured by true positive rate (TPR). **e** Precision measured by false discovery rate (FDR). The nominal FDR level of 0.05 for all methods (except scCODA with FDR 0.2) is indicated with a horizontal black line.

We understand that the results obtained from scCODA are crucial in this paper and we have provided an extended explanation about how scCODA works both in the results section

To investigate... *scCODA, a recently developed tool that can reliably identify changes in single-cell datasets even with a low number of replicates*⁵. In contrast to other methods, *scCODA models compositional bias of the sample as a whole and not for each cluster independently, which prevents identifying cell proportion changes due, for instance, to the depletion of a single-cell population. To identify compositional changes, we chose as reference group Fresh samples, so that the results will indicate if we find compositional changes relative to this group.*

and in the methods section

To run scCODA we defined fresh samples as a reference condition. To set the comparison, a cluster with low variability across samples had to be chosen as a reference. In this case we used as reference the NPC cluster, which had a good number of cells and a very low amount of dispersion

5 – Fig6: The authors argue that ACME and vivoPHIX cluster together but there is no statistical evidence that this is a significant or meaningful observation. In fact, all samples occupy the same area on a 2D space (e.g. Fig4) and can be readily integrated using Harmony, so it's not clear whether this would affect any downstream analysis at all. Additionally, an MDS or PCA plot of all pseudobulked samples would be more intuitive to read than the heatmap in Fig6.

We agree with the reviewer that this was not clearly done. As it is shown in Figure S3, before data integration, Methanol and Fresh samples occupy the same space in a joint UMAP while DMSO, ACME and VivoPHIX samples do not always overlap with them. This already indicates that there are changes in gene expression among the top 2000 variable genes that distinguish the samples by fixation method (and not by cell type). While the samples can be easily integrated with Harmony, which corrects the PCs considering the differences across samples, Harmony does not modify the expression of the genes included in the object (i.e. they have the same biases as before) and thus any expression biases depending on the fixation are still there.

Following the reviewer suggestion, we have determined the per cluster similarity among all samples using pseudobulk expression values. For each cell cluster, we have calculated the gene expression correlation across all samples and used this metric to perform a hierarchical clustering. We have used the method sigclust2 (Kimes et al. 2017) to investigate if the clustering of samples at the cell type level was significant or not. As can be seen in the new Figure 5 (see below), in all cases methanol samples cluster with fresh samples. In contrast, in 8 of the 12 cell populations identified most vivoPHIX and ACME samples cluster separately from Fresh samples.

We believe this analysis better supports our original claim. We have modified the text in the manuscript and the methods section describing this new analysis. The whole section discussing how fixation methods affect gene expression and sample composition has been rewritten to improve clarity and include the new analysis.

New Figure 5 caption: analysis identifies fixation-related biases in cell clustering. Dendrogram showing the similarity of the gene expression profiles of the cells from the different samples belonging to the same cell cluster. The samples are labelled as follows: fresh (F1 and F2, red color), methanol (M1, M2, M3 and M4, green color), DMSO (D1, D2 and D3, blue color), ACME (A1 and A2, purple color) and vivoPHIX (V1 and V2, magenta color). The significance in the hierarchical clustering is assessed using sigclust2. Significant branches in the dendrogram are highlighted in pink and highlighted with * is the p-value is < 0.05, ** if the p-value is < 0.01 and *** if the p-value is < 0.001. Not-significant branches are highlighted in yellow and not-tested branches in blue and green.

6 – The authors make the valid point that both dissociation and fixation could differentially affect subtypes of cells. It would be powerful to show per cluster (i.e. per cell type) module scores or DEGs to identify potential differences in how sample processing or fixation affect different cell types. Tools like muscat (Bioconductor) can facilitate multi-sample, multi-condition, cluster-resolved differential gene expression using pseudobulk expression levels, which would address most of my points.

Following the reviewer's suggestion, we have used Muscat to identify per-cluster differentially expressed genes depending on the fixation method. The results are included in the new Figure 6c and Table S6).

7 –There is extensive variability in how long the samples were preserved for before sequencing. It would be important to show whether the time preserved correlates with the QC readouts used in this MS – i.e. what changes in QC are due to the fixation vs the time the cells spent frozen down.

It is true that the preservation time affects the samples, as for instance, we observed that ACME sample1 (A1), which was preserved for 120 days, has many more cells discarded (Table 1) than the ACME sample2 (A2), and also lower UMI and genes. Yet, this observation is not consistent across fixation methods. For instance, VivopHIX sample 2 (V2), which was preserved only for 9 days have more cells discarded than VivoPHIX sample 1 (V1) (see Table). Also, we see that the methanol samples that were preserved for longer time (M3 & M4), have higher genes and UMIs detected and lower Ribosomal content than samples M1 & M2 that were preserved only for a week (see Figure R3 below).

Figure R3. Violin plots showing the distribution of the number of genes, UMIs, mitochondrial content (%MT) and Ribosomal content (% Ribo) of all the samples included in the analyses sorted by the number of days the samples were preserved, from 0 (D0) to 120 (D120) days. The data is the same data included in Figure 2c.

As described above, now we have included a “Limitations of this Study” section (see above) in which we also discuss the fact that other variables apart from fixation/preservation can affect sample quality.

8 – Another limitation of this study is that it is not assessing whether different fixation methods affect DEG analysis across different biological conditions (e.g. different culture/maturation conditions or drug treatments). While meOH might preserve homeostatic gene expression, it is unclear whether it can recover state-dependent DEGs. This could be discussed in the context of other limitations mentioned above.

Previous studies have shown that methanol fixed cells can be used to identify changes across biological conditions. For instance, a previous study used methanol fixed cells to assess the alterations in neurogenesis caused by point mutations in SURF1 gene, which is a model for Leigh Syndrome disease, at the single-cell level (Inak et al. 2021). Another study used Methanol fixed Macrophages to study the changes in their activation upon injury (Niehaus et al. 2021). Although none of these studies have compared the changes identified in fresh cells in comparison to methanol fixed cells, or using different fixation methods, they demonstrate that gene expression changes due to biological conditions are present in the samples.

Yet, considering that identification of DEG is often related with gene expression levels, it is likely that if fixation methods alter the detection power of genes, or their quantification, as we see in VivoPHIX and ACME cells, this could interfere with the detection of DEGs. We have included these thoughts in the discussion

Finally, it has to be considered that fixation methods could affect the detection of differentially expressed genes as they affect the library complexity and the number of Genes and UMI. Given that the ability to detect differentially expressed genes is directly related to the number of reads or UMIs assigned to a gene, it is possible that subtle differential expression changes due to biological or experimental conditions may be missed. We expect that these effects would be higher in ACME and vivoPHIX samples, which show lower genes and UMIs detected per cell (Figure 2c), although this can be compensated if more cells are sequenced. Methanol fixed cells have been successfully used before to discover changes in gene expression in single-cell data across different biological conditions^{37,38}, indicating that this fixation method is suitable not only to identify homeostatic gene expression but also to identify more subtle biological differences.

Reviewer #3 (Remarks to the Author):
scRNA-seq in neurons

In this manuscript, the authors compare different fixation methods of iPSC-derived neural precursors to determine which strategy is optimal for future scRNA-seq experiments on a droplet-based platform. A detailed comparison is a welcome addition to the field, as the authors correctly note that distinct fixation protocols can have significant differences on the preservation and quality of RNA integrity and scRNA-seq results. Notably, neurons present an additional challenge as they can be more sensitive to fixation, freezing and thawing methods, with many reagents on the market dedicated to preserving neural precursors and mature neurons. While the authors do note some important and striking differences between the different cell preservation strategies, their lack of detail in their methodology complicates my interpretation of some of their findings and makes it

difficult to determine how much of the variability is due to the specific fixation reagents vs. other factors the authors may have overlooked.

My detailed comments are below:

1. In an ideal scenario, the authors would have generated a large batch of iPSC-derived neurons and then performed multiple fixation protocols on this same batch of cells (possibly being done 1-2 days apart if necessary to minimize complications of performing different freezing experiments on the same day). This strategy would minimize/eliminate any batch-to-batch variability because different freezing conditions would be done on the exact same cultures and thus any differences would almost certainly be due to freezing/thawing preparations. Is this how the authors performed their experiments? Or instead, were all of the authors samples (F1, F2, M1, etc.) performed on separate differentiations?

We agree with the reviewer that the ideal scenario would be one including only 1 biological experiment. However, this is not the case. Most of the samples come from different differentiations, except M3, M4, D1 and D2, which all correspond to the same differentiation. This information has been clarified in the methods section and included in Table S1.

If each freezing experiment was a distinct differentiation, then I'm concerned about how inherent batch-to-batch variability between distinct differentiations could affect their conclusions. Please expand on the methodology used, and if each experiment was a distinct differentiation, then the authors must address this batch-to-batch variability and clearly describe how this affects interpretation of their results.

We understand the reviewer's concern and agree that batch to batch biological variability could affect our results. Yet, the fact that the results performed with scCODA are significant despite the variation on the cell lines used (Figure 4b and Figure S6) indicates that this result is robust. We have also included a section termed "limitations of the study" in which we comment about the effects of batch to batch variability in our results.

For example, M3 and D1 (and M4 and D2) have identical differentiation days (40 or 50, respectively), preservation days (60) and cell lines used (AD5), all from Table S1. Were M3 and D1 from identical differentiations, and similarly were M4 and D2 from the same differentiation?

Yes

If so, the authors should clearly state this, as it makes the differences between DMSO and Methanol conditions clearly dependent on the fixation conditions and not due to batch-to-batch variability.

Following the reviewer's suggestion, we have included the following text in the discussion.

Different cell lines and experiments have been used for this project, which could be a confounding effect affecting cell type composition. However, the comparison of DMSO and Methanol samples that come from the exact same differentiation (M3, M4, D1 and D2) (Table S1) highlights a clear difference in the relative abundance of excitatory neurons population between methanol and DMSO samples, providing additional

evidence that the compositional biases are due to fixation/preservation procedure (Figure S6).

2. Another important methodological distinction: RNase inhibitor and DTT was included in the Methanol, ACME and vivoPhix preparations, but not in the DMSO preparation. This seems like a pretty important point and a potential (I'd argue likely) causative reason for differences between DMSO and Methanol preps. Do the authors have a specific reason for excluding RNase inhibitor and DTT from the DMSO fixation (i.e., do they not dissolve in the DMSO fixative)? Do the authors think this could contribute to the significant loss of postmitotic neurons in the DMSO fixation procedure? I would really like to see a sample where RNase and DTT are included in the DMSO fixation to have a better idea how important these reagents are in fixation process.

DMSO is not

DMSO is not a fixative compound but a cryoprotectant commonly used for cell banking purposes as it prevents the formation of crystals during the freezing process (see for instance Whaley et al 2020; PMID 33757335). Cells fixed in Methanol, ACME and VivoPHIX are permeabilized and that facilitates the degradation of RNA by RNases that can enter the cell. The addition of RNase inhibitors is done to inactivate ribonuclease enzymes and prevent the degradation of RNA inside cells. DTT is added to reduce disulfide bonds in RNases needed by RNases for stability, thereby inhibiting RNase activity and preserving mRNA for the RT reaction (Chen et al. 2004; PMID: 15604666). Taking this into account, the addition of DTT and RNase inhibitors on DMSO cryopreserved cells is not necessary.

On the other side, despite the broad use of DMSO as cryoprotectant, previous studies have reported effects of DMSO in multiple cell types (Awan et al. 2020; PMID: 32342730). In this paper, we have not tested if post-mitotic neural loss is due to the freezing/thawing cycling or to the DMSO toxicity on neurons. Yet, both effects could contribute to the reduction in the number of post-mitotic neurons in DMSO cryopreserved samples.

3. Another confusing thing about the methods description was the variability in 'days of differentiation' and 'days preserved' between the samples. How different is the cell composition (and gene expression) in samples harvested at differentiation day 29 (V1) compared to differentiation day 50 (M4&D2)?

We apologize for the confusion. The *days of differentiation* refers to the days that the cells have been in culture since the start of the neural induction, and this varies between 29 and 50 days. The *days preserved* refers to the number of days cells have been frozen before thawing, rehydrating and encapsulating them. As the reviewer noted, we expect changes in cell composition during the differentiation. According to the original protocol (Shi et al. 2012), we expect mature neurons to arise ~60. Considering that not all cells mature at the same speed, and that each differentiation is different, in general we would expect more mature neurons with more days of differentiation.

We have run scCODA to see if there were significant differences in cell composition using the days of differentiation as replicates instead of fixation methods. For this purpose, we used the NPC cluster from day 36 as a reference. As can be seen in the figure below (Figure R4), we do not find significant differences in sample composition due to

the number of days of differentiation. This does not mean that there are no differences, but rather that the differences between fixation/preservation methods, which is what we are interested in, are stronger.

Figure R4. Barplot showing the average proportion of cells per cluster when considering the days of differentiation instead of the preservation method. The samples are group as follows: D29 (V1), D36 (M1, M2, A1, A2), D40 (F2, M3, D1, V2), D44 (F1), D50 (M4, D2).

Couldn't the difference in differentiation days partially explain for some differences in cell types detected?

It's possible that the days of differentiation, or the cell lines used, affect the number of cell types detected. Yet, the strong depletion of neurons observed in DMSO samples cannot be explained by the number of days of differentiation. Considering that the cell composition analysis compares each fixation method to fresh, if anything, we would expect DMSO samples (that have been 40, 50 and 39 days under differentiation) to have the same proportion of neurons as fresh samples (41 and 44 days in culture respectively). The fact that the days of differentiation can affect the cell proportions have been included in the *limitations of this study* section.

The authors should discuss this issue in more detail. Similarly, do the authors believe that there are any differences between samples preserved for 7-9 days vs. 120 days? They could generate some insights into this by directly comparing A1 (120 days) vs. A2 (7 days), although it looks like these two samples were generated with different cell lines.

As stated above (see comments to reviewer 2 and Figure R3), it is true that the preservation time affects the samples, as for instance, we observed that ACME sample1 (A1), which was preserved for 120 days, has many more cells discarded (Table 1) than the ACME sample2 (A2), and also lower UMI and genes. Yet, this observation is not consistent across fixation methods as in other cases we see the opposite behavior. Given that we have not done an exhaustive comparison of how preservation days affects sample quality, we have included this on the "limitations of the study" section.

4. The authors appear to use 2 iPSC lines (Ctrl and AD), but they don't mention these or describe them in the Methods. Are there differences between these 2 cell lines, do they generate the same proportion of progenitors and neurons, etc.

In the paper we use 6 different iPSC cell lines derived from old healthy subjects (Rodríguez-Traver et al. 2020; PMID: 31794941) or sporadic AD patients with no known risk alleles (Díaz-Guerra et al. 2019; PMID: 31401456). This information is provided in Table S1 along with other sample information such differentiation days, preservation days, cell viability. As no single-cell transcriptomics has been previously done, we do not know if all cell lines generate the same proportions of cells.

Several sample preparations use only 'AD' iPSCs (DMSO and CellCover) whereas other conditions use both 'Ctrl' and 'AD' iPSCs (Fresh, Methanol, ACME, VivoPHIX). Combining these different conditions complicates interpretation of their data. It might be better to focus solely on AD5 cells since they were used for all sample preparations. For example, I would like to see comparative analysis specifically between M3, M4, D1 and D2, as they all use the same cell line and other factors. Adding M1 and M2 to the Methanol analysis, which were frozen for only 7 days, complicates a direct comparison between the Methanol and DMSO conditions.

We agree with the reviewer that the use of several iPSC cell lines could difficult the interpretation of results. To check that the compositional bias in DMSO samples was not due to the fact that both samples were generated with the same iPSC cell line (IC-AD5-F-iPS-4F-1), now we have included an additional sample (D3) generated with a pool of cell lines (IC-Ctrl1-F-iPS-4F-1, IC-Ctrl2-F-iPS-4F-1, IC-Ctrl3-F-iPS-4F-1), so that all the fixation/preservation methods we use at least 2 different cell lines and two different differentiation experiments. This is the same pool of cell lines used in M1, A1, and V1 samples. As can be seen in Figure R2/figure S6, the reduced amount of excitatory neurons cell populations (magenta boxes) in DMSO samples cannot be due to the cell lines used. D1 and D2 used the same cell line as F2, M3 and M4 samples and D3 the same cell line pool as M1, A1, and V1. In both cases, all other samples have more excitatory neurons than DMSO cryopreserved samples.

In any case, we discuss the limitations of the study considering all possible confounding factors in the new *limitations of this study* section.

5. In Fig S3, the cells fixed with DMSO show distinct, almost non-overlapping profiles compared to the other 4 conditions. I'm not sure why these dot plots are a different shape from the ones in Figure 3-5, but it would be nice if they were from the same plot so that I could know the different cell types (progenitors vs. postmitotic cells).

The plots in Figure S3 are done before sample integration with Harmony and their purpose is to show the batch effects present in the sample and the need for sample integration to compare the samples. We have clarified this in the text.

As can be seen in Figure S3, before integration the cells from different experiments occupy different regions of the UMAP (Figure S3).

Do the authors know why the DMSO cells are completely separated from the other conditions on this UMAP plot while they are overlap much better in Figures 3-5? Was

anything computationally different between processing the data in Fig S3 vs. Figs 3-5? And the authors should reference Fig S3 when discussing that DMSO is known to be toxic to the cells and thus could be causing significant transcriptional changes in the cells.

We are sorry about the confusion. We have clarified the difference between these plots and the integrated UMAP in the text.

REVIEWERS' COMMENTS:

Reviewer #1 (Remarks to the Author):

The authors have sufficiently addressed my concerns. I now support publication of the manuscript.

Reviewer #2 (Remarks to the Author):

The authors have addressed all my concerns.

Reviewer #3 (Remarks to the Author):

The authors have done a very good job addressing all of the reviewers' comments. The addition of new analysis strengthens some of the authors original claims. And additional details in their methodology clarifies some of my concerns. The authors' discussion about caveats and alternate sources of variability is welcome, as it should be good guidance for readers as they pursue different fixation strategies for their samples. This is a much improved manuscript.